# Decreased but diverse activity of cortical and thalamic neurons in consciousness-impairing rodent absence seizures

Cian McCafferty[1,2], Benjamin F. Gruenbaum [1], Renee Tung [1], Jing-Jing Li[1], Xinyuan Zheng [1], Peter Salvino[1], Peter Vincent[1], Zachary Kratochvil[1], Jun Hwan Ryu[1], Aya Khalaf[1], Kohl Swift[1], Rashid Akbari[1], Wasif Islam[1], Prince Antwi[1], Emily A. Johnson[1], Petr Vitkovskiy[1], James Sampognaro[1], Isaac G. Freedman [1], Adam Kundishora [1], Antoine Depaulis[3], François David[4], Vincenzo Crunelli [4], Basavaraju G. Sanganahalli [5,6,7], Peter Herman [5,6,7], Fahmeed Hyder [5,6,7] & Hal Blumenfeld [1,5,8,9] ✉

Absence seizures are brief episodes of impaired consciousness, behavioral arrest, and unresponsiveness, with yet-unknown neuronal mechanisms. Here we report that an awake female rat model recapitulates the behavioral, electroencephalographic, and cortical functional magnetic resonance imaging characteristics of human absence seizures. Neuronally, seizures feature overall decreased but rhythmic firing of neurons in cortex and thalamus. Individual cortical and thalamic neurons express one of four distinct patterns of seizure-associated activity, one of which causes a transient initial peak in overall firing at seizure onset, and another which drives sustained decreases in overall firing. 40–60 s before seizure onset there begins a decline in low frequency electroencephalographic activity, neuronal firing, and behavior, but an increase in higher frequency electroencephalography and rhythmicity of neuronal firing. Our findings demonstrate that prolonged brain state changes precede consciousness-impairing seizures, and that during seizures distinct functional groups of cortical and thalamic neurons produce an overall transient firing increase followed by a sustained firing decrease, and increased rhythmicity.

Absence epilepsy causes recurring brief episodes of impaired consciousness. The altered consciousness in absence seizures can be highly disabling, characterized by behavioral arrest and unresponsiveness[1,2]. Electroencephalography (EEG) during absence seizures reveals abnormal paroxysmal brain rhythms called spike-wave discharges (SWDs) associated with behavioral changes. However, the

way that SWDs disrupt normal neuronal activity to produce unconsciousness is not known. It is increasingly recognized that absence epilepsy is not a benign condition and that long-term outcomes are variable[3–6]. This variability, including in the efficacy of gold-standard pharmacological interventions[5], suggests the need for targeted therapeutics that would in turn require the identification of molecular or

[1]Department of Neurology, Yale School of Medicine, New Haven, CT 06520, USA. [2]Department of Anatomy & Neuroscience, University College Cork, Cork, Ireland. [3]Univ. Grenoble Alpes, Inserm, U1216, Grenoble Institut Neurosciences, 38000 Grenoble, France. [4]Neuroscience Division, School of Bioscience, Cardiff University, Cardiff, UK. [5]Magnetic Resonance Research Center, Yale University, New Haven, CT 06520, USA. [6]Department of Radiology and Biomedical Imaging, Yale University, New Haven, CT 06520, USA. [7]Department of Biomedical Engineering, Yale University, New Haven, CT 06520, USA. [8]Department of Neuroscience, Yale School of Medicine, New Haven, CT 06520, USA. [9]Department of Neurosurgery, Yale School of Medicine, New Haven, CT 06520, USA. ✉e-mail: hal.blumenfeld@yale.edu

cellular mechanistic targets. Therefore, understanding the nature of neuronal activity producing impaired consciousness in absence seizures has important translational value, including changes in the overall level of neuronal firing during SWDs with impaired consciousness and, more specifically, whether absence seizures reflect activity in all neurons the same way or whether different neuronal subpopulations may contribute to absences via distinct activity patterns. Finally, the longer timescale state changes in neuronal activity leading to seizure initiation are unknown.

Animal models of absence epilepsy have provided insights into the pathophysiology of generalized SWDs in cortical and thalamic networks[7–12]. Meanwhile, human studies have characterized in detail the behavioral aspects of impaired consciousness in absence seizures, including loss of responses to external stimuli and a pause in ongoing interactive behaviors[13–18]. Human research has also successfully related the impaired consciousness in absence seizures to measurements using functional magnetic resonance imaging (fMRI), showing that the magnitudes of fMRI changes in both cortex and thalamus are associated with the severity of impaired behavioral responses[14,19,20]. Specifically, the cortex in human fMRI studies of absence seizures shows mainly fMRI decreases whereas the thalamus shows fMRI increases[14,20–24]. Investigating the neuronal basis of these changes in a valid animal model could provide important further mechanistic insights. Evaluating consciousness in human and animal model epilepsy research in particular typically involves testing responsiveness and/or ability to report experiences[25] as particular indicators of consciousness[26] that have direct relevance for the impact of an epileptic seizure on an individual's safety, wellbeing, and ability to interact with their environment. In this study we therefore consider impaired consciousness in this context, focusing on tests of responsiveness more easily assessed in animal models.

However, the ability to link animal model data to human absence and to thereby explain the impaired consciousness on a neuronal level has been hampered by several major obstacles. First, despite many human studies showing impaired responsiveness and behavioral arrest during absence seizures, no prior studies have reproduced these findings in an animal model of absence epilepsy, in part because such studies[27,28] have tended to focus on the arrest of spontaneous behavior rather than the impairment of engaged behavior. Rat and mouse absence epilepsy models have face validity in terms of developmental features, comorbidities, and response of SWDs to pharmacological agents[29–31]. However, the lack of fully characterized behavioral impairment during rodent SWDs has impeded progress, and it has even been proposed that SWDs in rodents might represent a normal activity pattern[32,33]; but see also ref. 34. The translational value of rodent absence epilepsy models would be greatly enhanced by convincing evidence of behavioral impairment during SWDs similar to the human disorder.

Another important obstacle in relating animal model neuronal activity to the changes in human cortical and thalamic networks has been the lack of agreement between fMRI changes in human and animal model SWDs. Human fMRI studies consistently show predominantly decreased fMRI signals in the cortex during absence seizures, raising the intriguing possibility that impaired consciousness is related to overall depressed cortical neuronal function[14]. Seizures are commonly thought of as pathologically increased neuronal activity, so depressed cortical activity as the main hallmark of absence seizures would be a departure from the conceptual basis of other seizure types. However, in contradiction to human fMRI, numerous animal model studies have shown fMRI increases in the cortex during SWDs[35–38]. This discrepancy may be because all prior animal fMRI studies were done under some form of anesthesia or sedation, or because of a neuronal or hemodynamic difference between animal SWDs and human absence seizures. Prior work has not yet investigated both fMRI changes and direct neuronal

recordings under the same conditions in an awake undrugged animal absence epilepsy model.

To overcome these obstacles, we sought to replicate human neuroimaging findings in absence epilepsy by developing a method using awake, restraint-habituated genetic absence epilepsy rats of Strasbourg (GAERS), thereby eliminating the potential confound of anesthetic agents used in prior fMRI studies. Anesthetic drugs can abnormally enhance neuronal excitability in absence models[35]. Prior work in GAERS has demonstrated seizure-related rhythmic activity in cortical and thalamic neurons but has not investigated fMRI and neuronal changes without anesthetic drugs[8,10,39]. With our protocol we were able to obtain fMRI measurements in GAERS during SWDs in the unanesthetized state, which yielded findings closely resembling human absence. Replicating the testing of behavior during SWDs in animal models has been challenging because seizures are more likely to occur in a state of relaxed wakefulness and tend to be entirely suppressed during active engagement in a task[28,40,41]. As such, previous studies of animal SWDs have not successfully employed conventional tests of consciousness, including ability to maintain ongoing interactive behaviors or responsiveness to external stimuli used in human studies[14,18,25]. In this study we devised modified paradigms to investigate behavioral continuity and responsiveness during SWDs in GAERS. We motivated rats to perform a repetitive motivated licking task by presenting them with unheralded rewards at irregular intervals. Separately, we developed a closed-loop test of auditory responsiveness by decreasing stimulus intensity and increasing interstimulus interval durations to achieve behavioral testing without SWD interruption.

Having established the neuroimaging and behavioral validity of the absence rodent model, we were then able to invasively record patterns of neuronal activity relevant to seizure-induced loss of consciousness. Using ensemble electrophysiological recordings with spike sorting, we showed that overall neuronal activity in both cortex and thalamus is decreased during these behavior-impairing SWDs. Interestingly, this overall decrease during SWDs is consistently driven by a functional subset of neurons in both cortex and thalamus, while the remaining neurons fall into one of three other seizure-associated activity patterns. One of these subsets appears to explain the transient peak in overall firing seen around the start of seizure. In addition, 2–3 s immediately preceding SWD onset there was a dip in neuronal firing across diverse cortical and thalamic neurons. Finally, we observed multi-modal changes preceding seizure initiation by tens of seconds. Approximately 40–60 s before seizures were detectable on the EEG, both power in high frequency bands and spontaneous behavior (licking) began to gradually decrease, while neuronal rhythmicity increased and firing rates decreased over similar timescales. This is also consistent with evidence of pre-ictal changes on longer timescales from human absence seizures[14,20,24,42,43]. These findings provide evidence, in a validated model of overall decreased cortical and thalamic neuronal activity, of distinct firing patterns in neuronal subgroups and of prolonged changes prior to seizure initiation, together giving insights into impaired consciousness and potential cell-specific mechanisms in absence seizures.

## Results
### fMRI during GAERS SWDs resembles human absence seizures
Human research in children with typical absence seizures has shown prominent fMRI decreases in most cortical regions during generalized SWDs and thalamic fMRI increases[14,16,19,20]. However, previous work in rodent models of absence epilepsy, done under anesthesia or other pharmacological intervention, has yielded variable results, with most studies showing cortical fMRI increases rather than decreases during SWD[36,37,39,44]. It was hypothesized that cortical fMRI increases in rodent models were produced by anesthetic and sedative agents not used in human studies; this hypothesis was further supported by observation

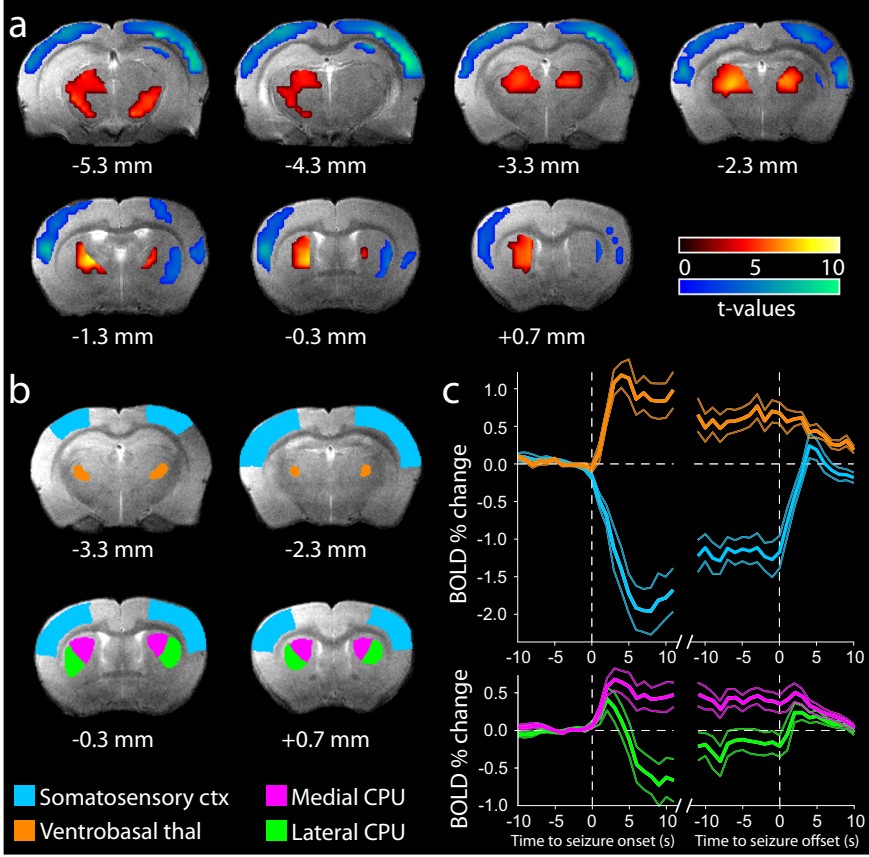

**Fig. 1 | BOLD fMRI signals associated with SWDs in GAERS resemble human absence epilepsy. a** Statistical Parametric Mapping (SPM) analysis of spike-wave discharge (SWD)-associated changes in blood-oxygen-level-dependent (BOLD) signal. Cortex shows mainly functional magnetic resonance imaging (fMRI) decreases (cool colors), whereas the thalamus shows fMRI increases (warm colors). Values on both color scales indicate the magnitude of increases (upper scale) and decreases (lower scale). T maps are superimposed on coronal anatomical images from the template animal, with FDR corrected threshold $p < 0.05$. AP coordinates are in millimeters relative to bregma[91]. **b** Regions of interest overlaid on a reference anatomical MRI image. Structures are taken from the Paxinos & Watson Rat Brain Atlas[91] after alignment of image sections with approximate rostrocaudal locations from bregma. Somatosensory cortex (ctx, cyan) includes all S1 regions from bregma +1 mm to bregma −3.6 mm; Ventrobasal thalamus (thal, orange) includes VPM and VPL from bregma −2.3 mm to bregma −4.16 mm. Note that these are representative sections only and do not constitute the full extent of the regions in question. **c** Mean percent-change time courses of BOLD signals (±SEM) in each of the regions described in **b** (including medial (pink) and lateral (green) caudate putamen (CPU)) aligned to SWD onset and offset in 1 s time bins. $N = 18$ animals, 670 SWDs in all of these analyses.

of fMRI increases in an anesthetized ferret model, despite the fact that the SWD had similar frequency to humans and the thalamocortical anatomy was more similar to humans than in rodent models[35]. We therefore sought to study neuroimaging signals in a rodent absence epilepsy model without anesthetic agents. The GAERS model was chosen due to its exclusive expression of typical absence seizures[12,45], the greater tractability of rats (vs. monogenic mouse models) to behavioral interrogation, and the etiological advantage of genetic vs. pharmacological models of absence seizures[12]. Drug-free neuroimaging was accomplished by gradual habituation of the GAERS to the ambient noise and cloth body restraint used in the neuroimaging environment, and use of carbon filament skull electrodes to allow simultaneous acquisition of EEG and blood-oxygen-level-dependent (BOLD) fMRI signals (Supplementary Fig. 1). In this awake model of spontaneous unanesthetized SWD, we found that fMRI signals from the cortex showed predominantly decreased activity consistent with human studies (Fig. 1)[14].

Specifically, both statistical parametric mapping (SPM) modeling using a seizure-associated boxcar convolved with the canonical hemodynamic response function (Fig. 1a), and region-of-interest (ROI) time courses of the percentage change in the BOLD fMRI signal (Fig. 1b, c) demonstrated decreases in the somatosensory cortex associated with seizures. Mean decrease in fMRI signal in the somatosensory cortex during SWDs (Fig. 1c) was $0.97 \pm 0.18\%$ SEM ($p = 4.3 \times 10^{-5}$). These analyses also revealed a seizure-associated hemodynamic increase in the ventrobasal thalamus (mean increase $0.51 \pm 0.11\%$, $p = 0.0002$), and seizure-associated changes in the caudate putamen (increases predominating medially and decreases laterally).

The unanesthetized GAERS model therefore shows cortical fMRI signal decreases and thalamic increases during SWDs, similar to the changes associated with human absence seizures. The magnitude of the changes, at ~1%, are comparable to the ~0.5% changes observed in humans[14,20] and at the upper end of the general 0–1% expected range for normal (non-noise-induced) fMRI signal dynamics across species[46,47]. The relatively large changes in GAERS may be due to their selective breeding for the expression of robust, widespread SWDs.

### Impaired behavior during GAERS SWDs resembles human absence seizures

We investigated the behavioral consequences of GAERS SWDs using two different paradigms, designed to be analogous to the repetitive tapping task and continuous performance task previously employed to study human behavior during absence seizures[14,15,19]. It has been challenging in previous rodent absence model work to demonstrate impaired behavior during SWDs, at least in part because behavioral

tasks increase arousal which tends to suppress or interrupt SWDs, preventing testing of behavior during SWDs[28,48,49]. Therefore, in our studies both tasks were modified to maintain a level of engagement/arousal compatible with uninterrupted seizure expression. In one task, designed to echo the human continuous performance task[14], we initially conditioned GAERS to respond within 10 s to an 80 dB pure auditory tone at 8 kHz lasting 0.5 s, presented at 60 s intervals, in order to receive a sucrose water reward (Fig. 2a). Based on initial observations this sound intensity and stimulus interval maintained the rats in an aroused state and prevented the expression of SWDs. The interstimulus interval was therefore increased to a range of 150–210 s (or the automated detection of a SWD based on EEG amplitude, whichever came sooner) and the intensity decreased to 45 dB. Lowering stimulus intensity allowed SWDs to persist (for more than 1 s) after stimulus presentation in the majority of SWD episodes. To mirror studies done in human absence epilepsy[14,15,18], we analyzed behavioral responses only for SWDs that were not interrupted, so that response or lack of response to external stimuli during ongoing SWDs could be evaluated. During baseline periods between SWDs, animals had an $88.2 \pm 2.8\%$ response rate after conditioning (mean $\pm$ SEM, $n = 14$ rats with 4482 total stimuli). During SWDs the response rate decreased markedly to $0.4 \pm 0.3\%$ ($p = 0.00012$, Wilcoxon signed rank test; $n =$ the same 14 rats with 156 total stimuli). This behavior appeared to be restored immediately post-SWDs, with response rates recovering to $78.2 \pm 6.8\%$ ($p = 0.2163$ compared to pre-seizure, Wilcoxon signed rank test; $n =$ the same 14 rats with 330 total stimuli) (Fig. 2c, d).

In the second task, designed to resemble the human repetitive tapping task[14], the instructed repetitive tapping was approximated by providing unheralded sucrose rewards at varying intervals (Fig. 2b). This paradigm encouraged spontaneous licking at the spout at a mean rate of $0.54 \pm 0.07$ licks/second outside of SWDs (27 rats) and was therefore referred to as a sustained motivated licking task. We observed a decrease in lick rate from ~0.75 licks/second 10 s pre-SWD to a mean lick rate during seizures of $0.007 \pm 0.002$ licks/second ($n = 27$ rats, total of 3146 seizures), constituting a significant decrease from non-seizure periods ($p = 5.6 \times 10^{-6}$, Wilcoxon signed rank test). Within 2–3 s of seizure end there was an apparent recovery in lick rate, with a mean post-seizure rate of $0.55 \pm 0.06$ licks/second ($p = 0.8288$ relative to all non-seizure periods, Fig. 2e, f).

These results constitute a rigorous demonstration of consistently impaired behavioral interactions with the environment during SWDs in a rodent absence epilepsy model. This provides important face validity for the rodent model to investigate mechanisms of impaired behavioral interactions in human absence epilepsy. As further validation of the model, we sought to determine whether behavior might be spared in some SWDs. In human absence epilepsy, behavior may be spared during some seizures, especially in tasks that are less behaviorally demanding, such as the repetitive tapping task[14,15,19]. In addition, in human absence epilepsy, the SWDs accompanying spared behavior are significantly less physiologically severe in magnitude and duration[14,16]. Similarly, in the GAERS model we found that performance on the more demanding auditory response task was virtually always impaired during SWDs, whereas in the less demanding spontaneous licking task, ~5% of all SWDs (158/3146) demonstrated some persistent licking during SWDs. Behaviorally "spared" seizures included at least 1 lick during the seizure, whereas "impaired" seizures were defined as those with no licks (see Methods, Behavioral Data Acquisition with Simultaneous EEG and Behavioral Data Analysis section for details). Again resembling human absence seizures[14], the SWDs with spared behavior in the rodent model featured significantly lower EEG power in bands corresponding to both the wave (5–9 Hz, $p = 4.691 \times 10^{-8}$, rank sum test, Fig. 2g) and the spike (15–100 Hz, $p = 2.146 \times 10^{-7}$, rank sum test, Fig. 2h) components of the spike-wave oscillation, as well as lower overall power (see Supplementary Fig. 2B). Also similar to human

absence epilepsy[14], the mean duration of SWDs with spared behavior ($5.9 \pm 0.6$ s) was significantly shorter than that of SWDs with impaired behavior ($8.9 \pm 0.2$ s, $p = 8.8 \times 10^{-9}$, rank sum test, Supplementary Fig. 2A).

These behavioral results, in agreement with our fMRI hemodynamic observations, suggest that SWDs in GAERS have similar physiological changes in corticothalamic networks and similar behavioral effects on consciousness as do absence seizures in humans. As such, neuronal activity during these SWDs may provide valuable insight into potential mechanisms of absence seizures and their symptoms.

**Total neuronal activity accompanying GAERS SWDs is decreased**
Having established that GAERS SWDs were accompanied by impaired behavior, we investigated the changes in neuronal activity that might cause these impairments. Our investigation of neuronal activity was targeted at representative cortical and thalamic regions known to be involved in SWDs based on prior work. Cortical involvement in rodent SWD is most prominent in somatosensory cortex, particularly in the peri-oral areas[8,9]. Specifically, deep layer peri-oral somatosensory neurons appear to initiate SWDs in GAERS, and this region of cortex temporally leads SWD activity in the related WAG/Rij model. Because our focus was on general cortical activity rather than on focal seizure initiation, we therefore recorded from subgranular layers (0.9–2 mm below pial surface) of somatosensory cortex in the trunk region (Supplementary Fig. 3). In addition, to investigate thalamic activity during SWDs, we performed new analyses on previously acquired recordings of neuronal activity from ventral basal somatosensory thalamus[10]. First, we studied the firing of 168 individually sorted neurons in the somatosensory cortex around SWD initiation and during SWDs, obtaining sortable action potentials during seizures (Fig. 3). The firing rate of these cortical neurons decreased from $4.5 \pm 0.4$ spikes/second (mean $\pm$ SEM) before seizure initiation to $3.5 \pm 0.4$ spikes/second during the seizure ($p = 0.001$, paired $t$-test; Fig. 4a, b). Interestingly, there was an abrupt dip in mean neuronal firing 2–3 s before seizure onset and then a transient peak in firing just at the point of seizure initiation (Fig. 4a). This was followed by a sustained decrease in firing that lasted until seizure termination. A similar pattern of sustained decrease in firing with an early dip and then peak at seizure initiation was also observed in mean thalamic neuronal firing (decrease from $13.2 \pm 0.9$ spikes/second before seizure to $9.1 \pm 1.0$ spikes/second during, $p = 6 \times 10^{-12}$; Fig. 4c, d).

To investigate the mechanisms of decreased overall corticothalamic firing during SWDs, we examined the firing pattern through analysis of the regularity of intervals between consecutive action potentials. Specifically, we defined rhythmicity as the inverse of the coefficient of variation of interspike intervals, calculated in 2 s time bins. This demonstrated a significant increase in firing rhythmicity during SWDs compared to baseline in both cortex (rhythmicity index $0.85 \pm 0.05$ SWD, $0.53 \pm 0.02$ baseline, $p = 2.3 \times 10^{-9}$, paired $t$-test) and thalamus ($0.70 \pm 0.02$ SWD, $0.42 \pm 0.01$ baseline, $p = 8.9 \times 10^{-38}$, paired $t$-test; Fig. 4e, f, h, i).

Plots of the mean distribution of firing centered in time around the EEG spike (most extreme voltage value) in the spike-and-wave complex showed rhythmic oscillating periods of neuronal firing both above and below baseline (Fig. 4e, h). Peaks in action potential firing coincided with the EEG spike, and troughs in firing coincided with the wave of the SWD. We found that on average the decreased firing during the wave relative to baseline (trough deficit) was significantly greater than the increased firing during the spike (peak surplus), which explained the overall decrease in firing during SWD ($p = 0.006$ cortex; $p = 0.007$ thalamus; Fig. 4g, j).

Importantly, although the neuronal firing during SWDs fluctuates between peaks and troughs (Fig. 4e, h) the total change in firing over longer timescales depends on the mean firing rate averaged over spike-wave cycles. Thus, the mean change in firing rate during SWD

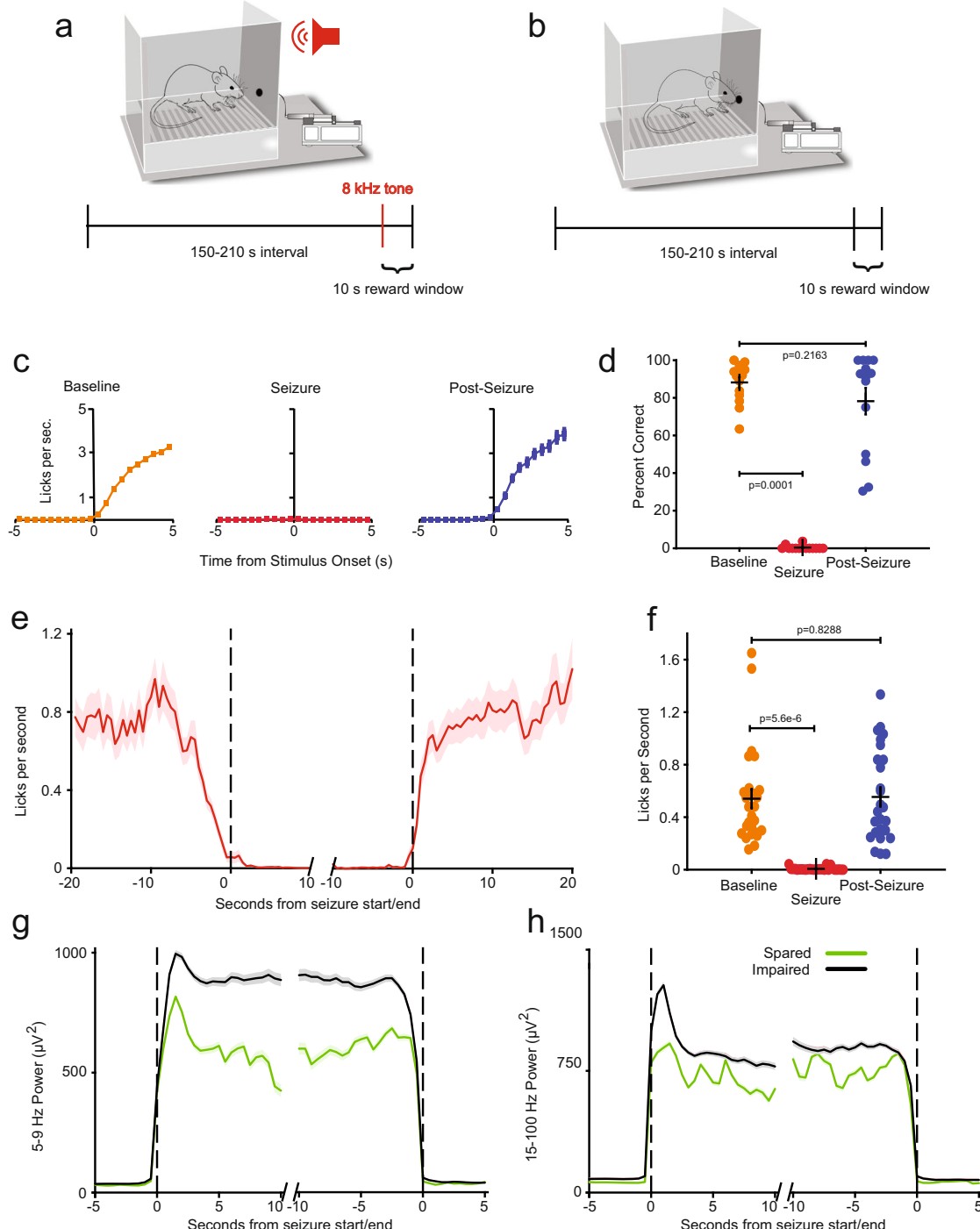

**Fig. 2 | Behavioral impairment of rats in two different tasks during absence seizures. a** Design of conditioned auditory response task ($n = 14$ animals (427 seizures)). **b** Design of continuous motivated licking paradigm ($n = 27$ animals (3146 seizures)). **c** Lick rate in 0.5 s bins following conditioned auditory stimuli delivered at baseline (amber, 20 s or more from the start and/or end of a seizure), during seizure (red), and in the 10 s immediately post-seizure (blue). ($n = 427$ seizures). **d** Percent of conditioned stimuli responded to within the reward window and within the specified state for baseline, seizure, and immediate post-seizure periods for each animal showing decrease during seizure and recovery in post-seizure ($n = 14$ animals, colors as in **c**). **e** Dynamics of lick rate in 0.5 s bins around seizure start and end times in continuous motivated licking paradigm ($n = 27$ animals). **f** Mean lick rates during baseline (20 s or more from the start and/or end of a seizure), seizure, and immediate (up to 20 s) post-seizure periods in

continuous motivated licking paradigm showing decrease during seizure and recovery in post-seizure ($n = 27$ animals, colors as in **c**). **g** Dynamics of electro-encephalogram (EEG) power in 1 s bins in the wave band (5–9 Hz) surrounding seizure start and end times for seizures with continued licking (green = spared, $n = 158$ seizures) and no licking (black = impaired, $n = 2988$ seizures). Wave power was significantly greater for impaired versus spared SWD ($p = 4.691 \times 10^{-8}$, rank sum test). See Methods for definitions of spared and impaired licking. **h** Dynamics of EEG power in 1 s bins in the spike band (15–100 Hz) around seizure start and end, comparing behaviorally spared ($n = 158$) and impaired ($n = 2988$) seizures. Spike power was significantly greater for impaired seizures ($p = 2.146 \times 10^{-7}$, rank sum test). For all plots (**c–h**), measures of center are mean, and error bars/regions denote standard error of the mean. Source data are provided in Source Data file.

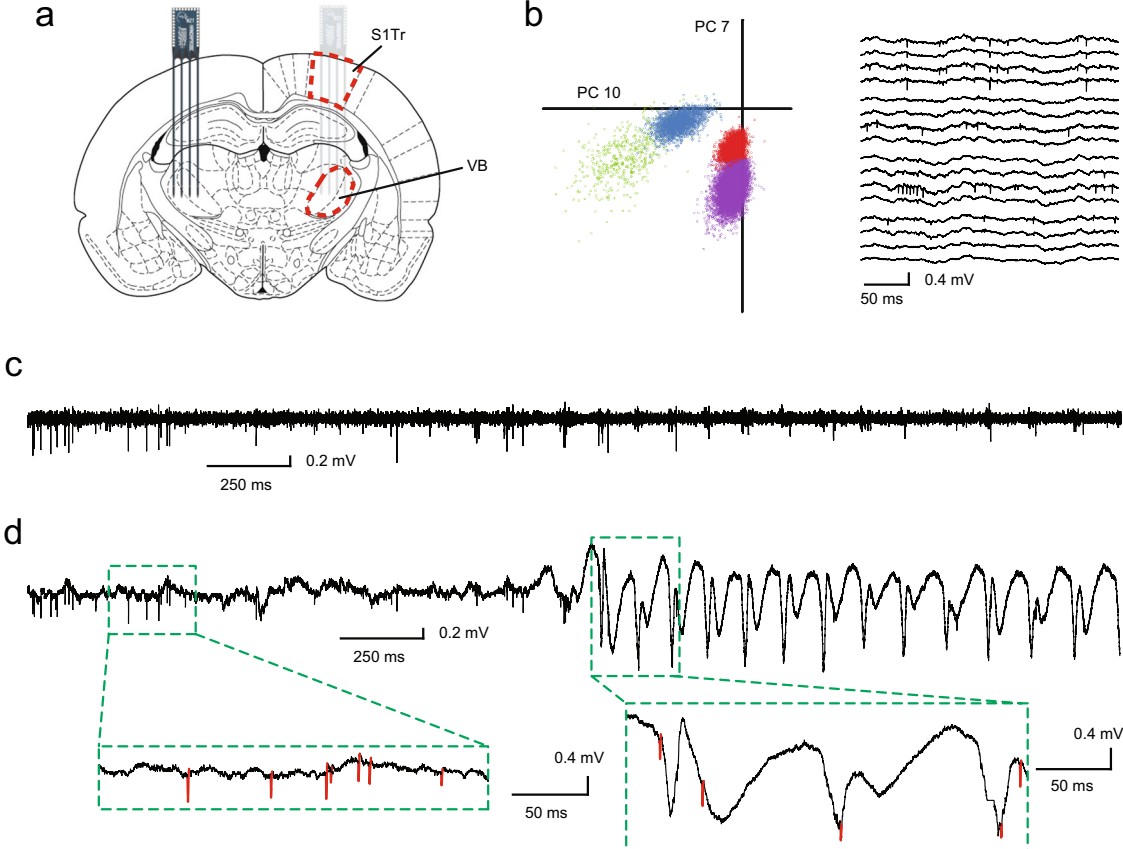

**Fig. 3 | Recording of individual neuronal action potentials in the cortex and thalamus of freely moving rats. a** Rat brain coronal section at bregma −3.6 mm (reproduced with permission from Paxinos & Watson Rat Brain Atlas[91]) indicating target locations of primary somatosensory trunk cortical (S1Tr) and ventrobasal thalamic (VB) electrode arrays. **b** Example principal component values of waveforms from simultaneously recorded cortical neurons (left) and sample raw broadband (1.1 Hz–7.6 kHz) voltage traces from a 16 channel array (right) from which neuron action potentials were extracted. Recording shows data during a non-seizure baseline period. PC7 and PC10 represent the first principal components from two channels in this example that show clear separation between clusters. **c** Single channel signal from same session as **b** but at a different time, during a transition from baseline waking to seizure state (see also part **d**). High-pass filtering by subtraction of median values with a 10-sample window shows identifiable action potential spike waveforms (negative deflections oriented downwards). **d** Raw (unfiltered) single channel data temporally aligned with trace **c** enables visualization of baseline waking state and spike-wave seizure activity, including action potential waveforms (red traces on insets).

compared to baseline can be expressed as follows:

$$\Delta S = (PS - TD) \times F \tag{1}$$

where $\Delta S$ is the mean change in firing rate (spikes/s), $PS$ is Peak Surplus (spikes/cycle), $TD$ is Trough Deficit (spikes/cycle), and $F$ is the SWD oscillation frequency (cycles/s). Therefore, if the Peak Surplus is smaller than Trough Deficit then the mean firing rate during SWD will decrease compared to baseline, regardless of the oscillation frequency.

These analyses indicate that overall cortical and thalamic neuronal activity during absence seizures is decreased, due to suppression of action potentials during waves exceeding excitation during spikes. We next asked whether the observed changes in total firing during SWD occur homogeneously in all cortical and thalamic neurons, or whether divergent firing patterns in different neurons may sum to produce the overall picture.

**Four classes of activity in different neurons during SWDs**
We considered whether the three primary features of neural firing during SWDs—an overall decrease in firing rate, a transient peak coinciding with SWD initiation, and an initial dip just prior to SWD onset—could be attributable to a subset of neurons. We found that the firing rate dynamics of cortical neurons around SWD initiation

tended to fall into one of four patterns (Fig. 5a upper). These were the following: a sustained decrease or increase in firing throughout the seizure (Sustained Decrease (SD) and Sustained Increase (SI) groups, 59 and 15 neurons; 37% and 9% respectively); a peak in firing at seizure initiation, followed by a return to baseline levels throughout (Onset Peak group (OP), 44 neurons, 28%); or no obvious change in firing rate associated with the seizure (No Change group (NC), 41 neurons, 26%). These patterns were distinctive and consistent within groups (extended data slides and single neuron raster plots are available at https://doi.org/10.5061/dryad.rfj6q57dz). Interestingly, analysis of thalamic unit activity around SWD revealed that the same four patterns of firing dynamics were apparent (Fig. 5a lower), with each neuron showing either a Sustained Decrease (76 neurons, 47%), Sustained Increase (7 neurons, 4%), Onset Peak (27 neurons, 17%), or No Change (52 neurons, 32%). Three cortical and one thalamic neuron did not fall into any of these categories and were not included in the analysis.

When quantifying changes in firing between the periods preceding and during SWDs, as for all neurons above, we observed that the SD groups were unsurprisingly responsible for the majority of the decrease in overall activity in both cortex and thalamus (Supplementary Fig. 4A, B). The NC group also exhibited smaller decreases in this comparison in thalamus only, while the OP group had no overall difference. The SI group had a statistically significant increase in cortex,

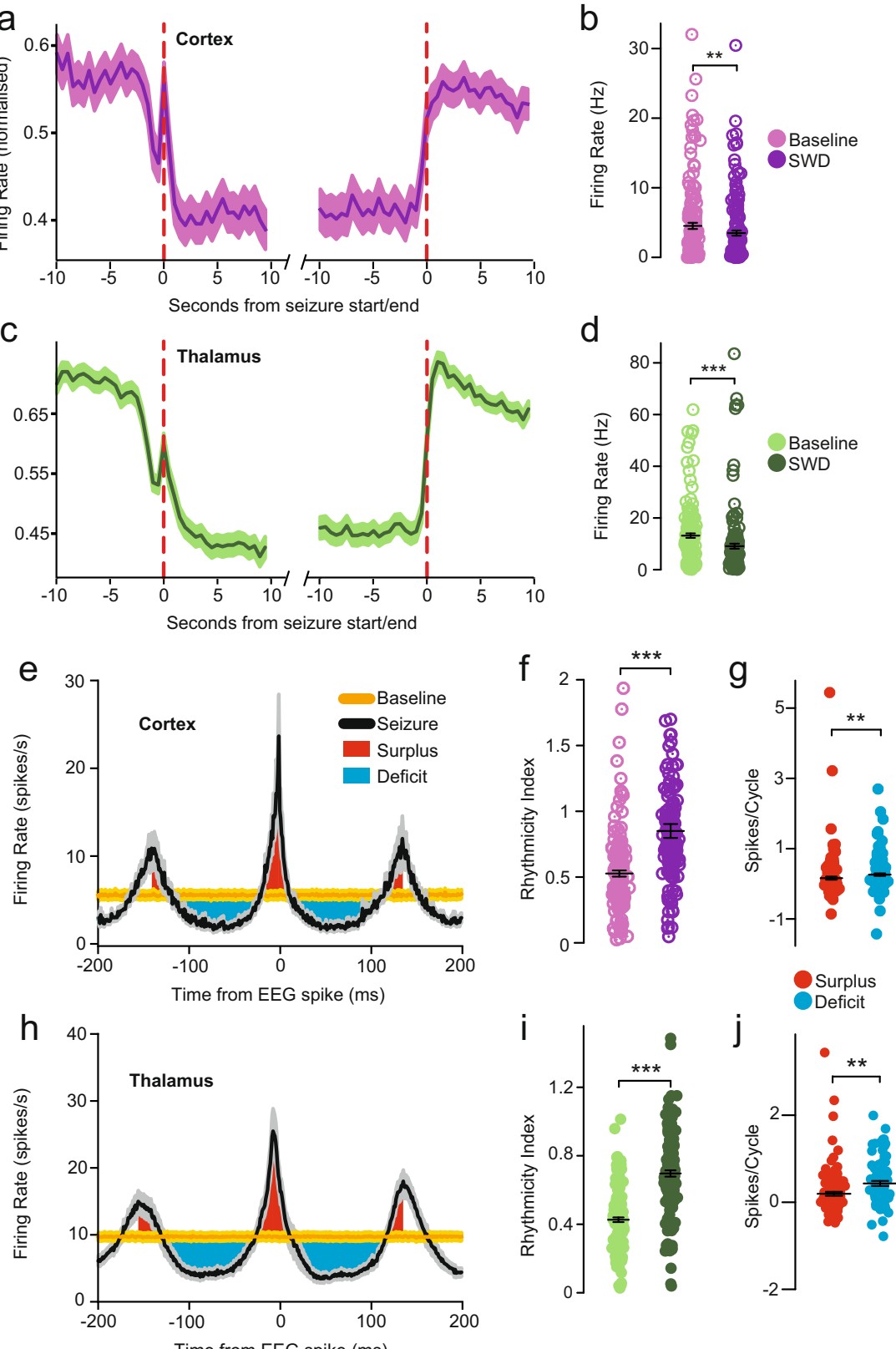

but not in thalamus (likely due to the small number of cells (7) in this group).

In contrast to this diversity of firing patterns in comparison to baseline across subgroups, all subgroups expressed a notable increase in rhythmicity of firing (defined as the reciprocal of the coefficient of variation of interspike intervals) during SWD compared to baseline periods (Fig. 5b, Supplementary Fig. 4C, D). The rhythmic alternation of firing, with relative increases during EEG spikes and decreases during waves, could explain the overall firing changes during SWD in each of the neuronal subgroups. As in the overall firing of total neurons

**Fig. 4 | Total firing of cortical and thalamic neurons during absence seizures shows decreased overall firing rate and increased rhythmicity. a** Mean firing dynamics around seizure start and end of all cortical neurons in 0.5 s bins, showing overall decrease in firing associated with seizures, as well as a dip in firing 2–3 s prior to seizure onset followed by a transient peak in firing that coincides with seizure initiation. **b** Mean firing rate in each cortical neuron showing overall lower firing rate during seizure than during 5 s pre-seizure baseline (*p* = 0.001). **c** Mean firing dynamics around seizure start and end of all thalamic neurons showing similar initial dip, transient peak and then overall decrease in firing. **d** As in **b** but for thalamic neurons (*p* = 5.4 × 10⁻¹²). **e** Distribution of action potentials around spike-and-wave complex electroencephalogram (EEG) spikes (time 0) in 1 ms bins of all cortical neurons, showing oscillations in firing that are relatively increased during the EEG spikes (Peak Surplus firing, red) and decreased during the EEG wave component (Trough Deficit firing, blue) relative to Baseline non-seizure periods (amber). **f** mean rhythmicity of each cortical neuron showing higher value during seizure than in 5 s pre-seizure baseline (*p* = 2.3 × 10⁻⁹). **g:** Mean absolute spike differential from baseline per spike-wave complex in peak and trough periods, for cortical neurons, showing that troughs exceed peaks (*p* = 0.006). **h** As in **e** but for thalamic neurons. **i** As in **f** but for thalamic neurons (*p* = 8.9 × 10⁻¹⁷). **j** As in **g** but for thalamic neurons (*p* = 0.0007). Data are from 165 cortical neurons in 7 animals and 163 thalamic neurons in 7 animals. All traces are mean and corresponding shading denotes standard error of the mean; all statistical tests are two-sided paired *t*-tests. Purple colors indicate cortical data (**a, b, f**) and green colors thalamic data (**c, d, i**). Source data are provided in Source Data file.

(Fig. 4g, j), we found that in the SD subgroup the decreased firing during the wave relative to baseline (trough deficit) was significantly greater than the increased firing during the spike (peak surplus), which explained the overall decrease in firing during SWD (Fig. 5b; see also Supplementary Fig. 4E, F). Conversely, in the SI subgroup the peak surplus exceeded the trough deficit (although not significantly so in thalamus, again due to low sample size; Fig. 5b; see also Supplementary Fig. 4E, F). The OP and NC groups showed no statistically significant differences between peak surpluses and trough deficits (Fig. 4b; see also Supplementary Fig. 4E, F).

Together, these results indicate that neurons consistently fall into one of four distinct patterns of activity around seizure start which all show increased rhythmicity during SWD, and that ~40–50% of all cortical and thalamic neurons are dominantly responsible for the sustained decrease in overall firing. The overall firing also showed an initial dip at the critical time 2–3 s before SWD initiation (Fig. 4a, c), which was reflected by a similar dip just before SWD onset in most cortical and thalamic neuron types (with the possible exception of thalamic SI and NC neurons, Fig. 5a). Interestingly, the peak in firing that coincides with SWD onset in the overall group activity (Fig. 4a, c), was observed to arise selectively from the OP neuron subset of cortical and thalamic neurons (Fig. 5a).

Because the transient surge in firing in a subset of neurons at SWD onset might play an important role in SWD initiation, we were interested in identifying the temporal characteristics that could explain the transient increased firing around the time of SWD onset in the OP subgroup. We therefore compared the properties of OP neuron firing in the first second after SWD onset versus the entire SWD (Fig. 5c, Supplementary Fig. 5). We found that OP neurons in both cortex and thalamus fired with higher oscillation frequency in the first second after SWD onset (cortex first second 8.5 ± 0.04 Hz, cortex entire SWD 7.2 ± 0.06 Hz, *p* = 4.5 × 10⁻¹⁶, Supplementary Fig. 5A; thalamus first second 7.9 ± 0.2 Hz, thalamus entire SWD 6.9 ± 0.1 Hz, 4.9 × 10⁻⁷, Supplementary Fig. 5C). However, this alone would not explain the unique transient peak feature in these neurons, as the higher first-second SWD frequency should affect the firing of all neurons, not just OP neurons. Therefore we investigated the intensity of firing of OP neurons during peak and trough periods in the first second of seizures. Both cortical and thalamic OP neurons had longer, larger peaks of firing yielding a greater peak surplus in the first second (cortex, *p* = 0.016; thalamus, *p* = 0.0005) (Fig. 5c; Supplementary Fig. 5B, D). Moreover, troughs of firing and consequent trough deficits were significantly smaller in the first second after SWD initiation in OP neurons (cortex, *p* = 0.007; thalamus, *p* = 0.005) (Fig. 5c; Supplementary Fig. 5B, D). Therefore, both differences in oscillation frequency, as well as differences in peak surplus and trough deficit firing patterns with each spike-wave oscillation cycle in the initial 1 s after SWD onset contribute to the transiently higher mean overall firing observed in the OP neurons (Fig. 5c). As per equation [1] above, a higher peak surplus relative to trough deficit, and a higher oscillation frequency will lead to a higher overall mean firing rate in OP neurons in the first 1 s after SWD onset.

## State changes occur tens of seconds prior to seizure onset

Previous work suggests that changes on longer timescales, including arousal state or other brain network changes, may occur prior to absence seizure initiation[20,50,51]. Therefore, we wished to investigate whether there were seizure-related changes in various parameters over longer pre-seizure periods (up to 120 s). The average spectrogram of frontoparietal EEG over this time period exhibited a marked decrease in high frequency (>40 Hz) and increase in low frequency (0–39 Hz) power over ~40–60 s prior to seizure initiation (Fig. 6a, b), suggesting a pre-seizure change in brain state.

We further investigated whether there were changes over a similar timescale in lick rates in the population of all seizures from our continuous motivated licking cohort of animals. The present analysis differed from Fig. 2, in which only seizures that had licking in the 10 s preceding initiation were considered (see Methods), whereas here we analyzed the licking rate preceding all seizures without exclusion. We found that seizures were preceded by a decrease in lick rate from a mean of 0.63 ± 0.02 to a mean of 0.04 ± 0.00 licks per second over ~40 s prior to SWD onset (Pearson's R = −0.9957, *p* = 1.17 × 10⁻⁸, Fig. 6c), a similar timescale to that of the electroencephalographic dynamics. Seizure termination was followed by a recovery to the baseline lick rate.

We next analyzed the role of our recorded neuronal populations in pre-seizure state changes. Cortical neurons appeared to decrease gradually in firing rate over a similar timescale to the other parameters, starting ~60–80 s pre-seizure (R = −0.5, *p* = 0.005, calculated from −80 to −10s pre-seizure, Fig. 6d upper). Further, their rhythmicity of firing increased over the same time period (R = 0.7, *p* = 1.9 × 10⁻⁶, Fig. 6d lower). Similar dynamics were observed in thalamic neuron firing (R = −0.8, *p* = 9.2 × 10⁻⁸) but not rhythmicity (R = 0.1, *p* = 0.56, Fig. 6e). These results suggest that neural and behavioral changes precede seizure initiation by tens of seconds, which with further work may provide important insights into the state of brain networks when seizures are initiated.

## Discussion

The present results demonstrate the neuroimaging and behavioral validity of the awake GAERS absence epilepsy model, and then reveal several interesting and important aspects of neuronal activity in absence seizures. We found that overall neuronal activity is reduced during seizures, that distinct firing patterns contribute to altered activity in different neuronal populations, and that long-term electrophysiological and behavioral changes begin 40–60 s prior to seizure onset. Prior work has not successfully replicated human fMRI and behavioral changes during absence seizures in an animal model, nor has it elucidated the neuronal activity patterns or the neuronal subtypes contributing to these changes. The present findings have important translational value, providing insights into mechanisms of impaired brain function and impaired consciousness during absence seizures.

We first validated the model by habituating GAERS to the MRI scanning environment, and demonstrated fMRI changes in the

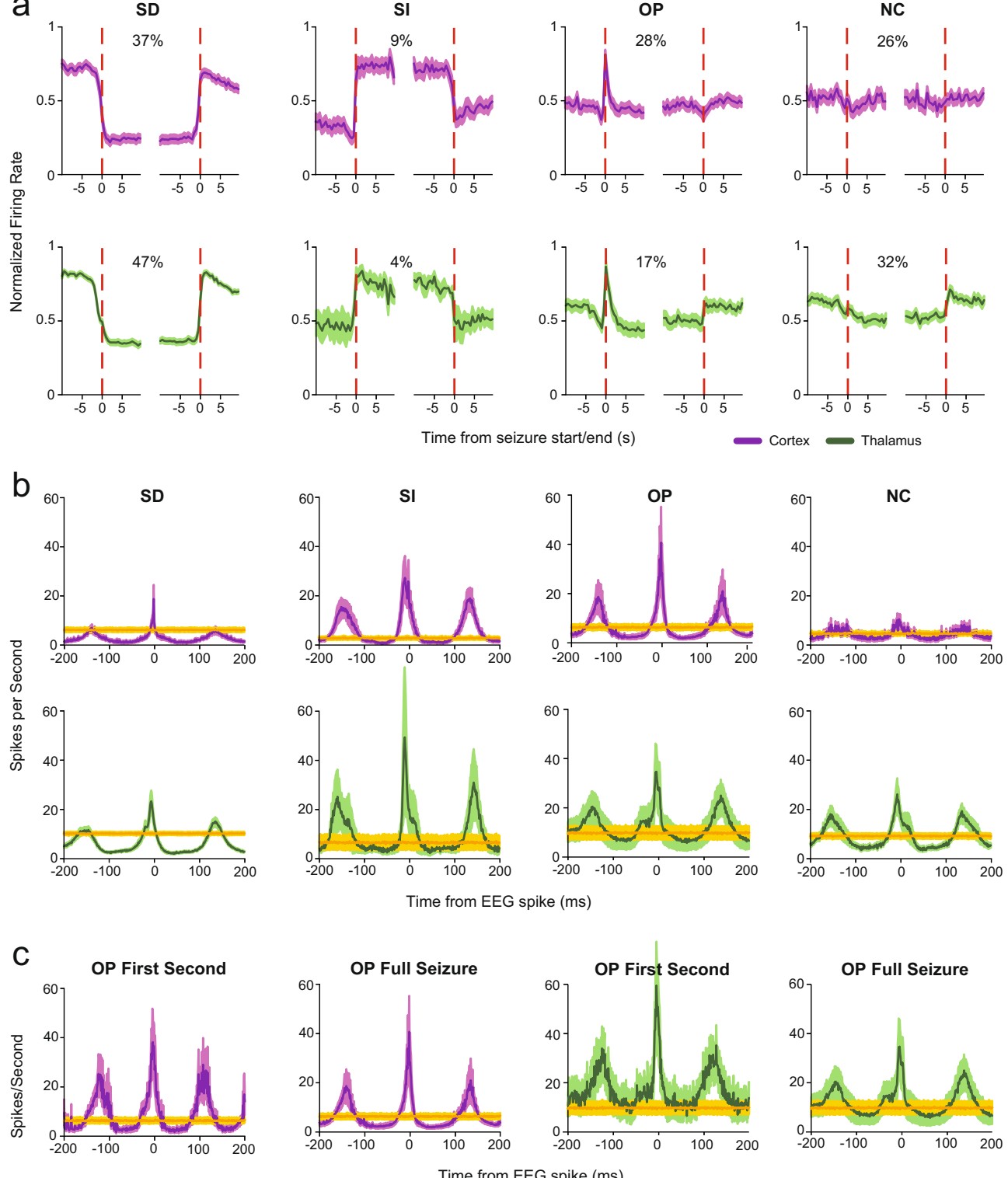

**Fig. 5 | Four distinct firing patterns of cortical and thalamic neurons during absence seizures. a** Firing rates in 0.5 s bins (scaled for each neuron relative to total range of firing in visualized period, see Methods) for each group of neurons showing the distinctive dynamics around seizure onset/offset from which their names are derived: SD sustained decrease (*n* neurons = 59 cortex, 76 thalamus), SI sustained increase (*n* neurons = 15 cortex, 7 thalamus), OP onset peak (*n* neurons = 44 cortex, 27 thalamus), NC no change (*n* neurons = 41 cortex, 52 thalamus). Also indicated is the percentage of all neurons in each group. **b** Mean distribution of action potentials in 1 ms bins around EEG spikes (time 0) of spike-wave cycles for the same groups. Oscillations within each cycle include periods of firing that are increased (peak surplus) or decreased (trough deficit) relative to baseline non-seizure periods (see also Fig. 3c; Supplementary Fig. 4e, f), which explain overall changes in firing seen during spike-wave discharges (SWD) for each group (see **a**). **c** Onset Peak neurons show higher oscillation frequency, and larger peaks in firing during first 1 s after seizure onset compared to the entire seizure period, explaining transiently increased firing at seizure onset. Data in **b** and **c** are from same neurons as in **a**. All traces are mean; shading denotes standard error of mean. Purple indicates cortical seizure data, green indicates thalamic seizure data, and amber indicates baseline firing.

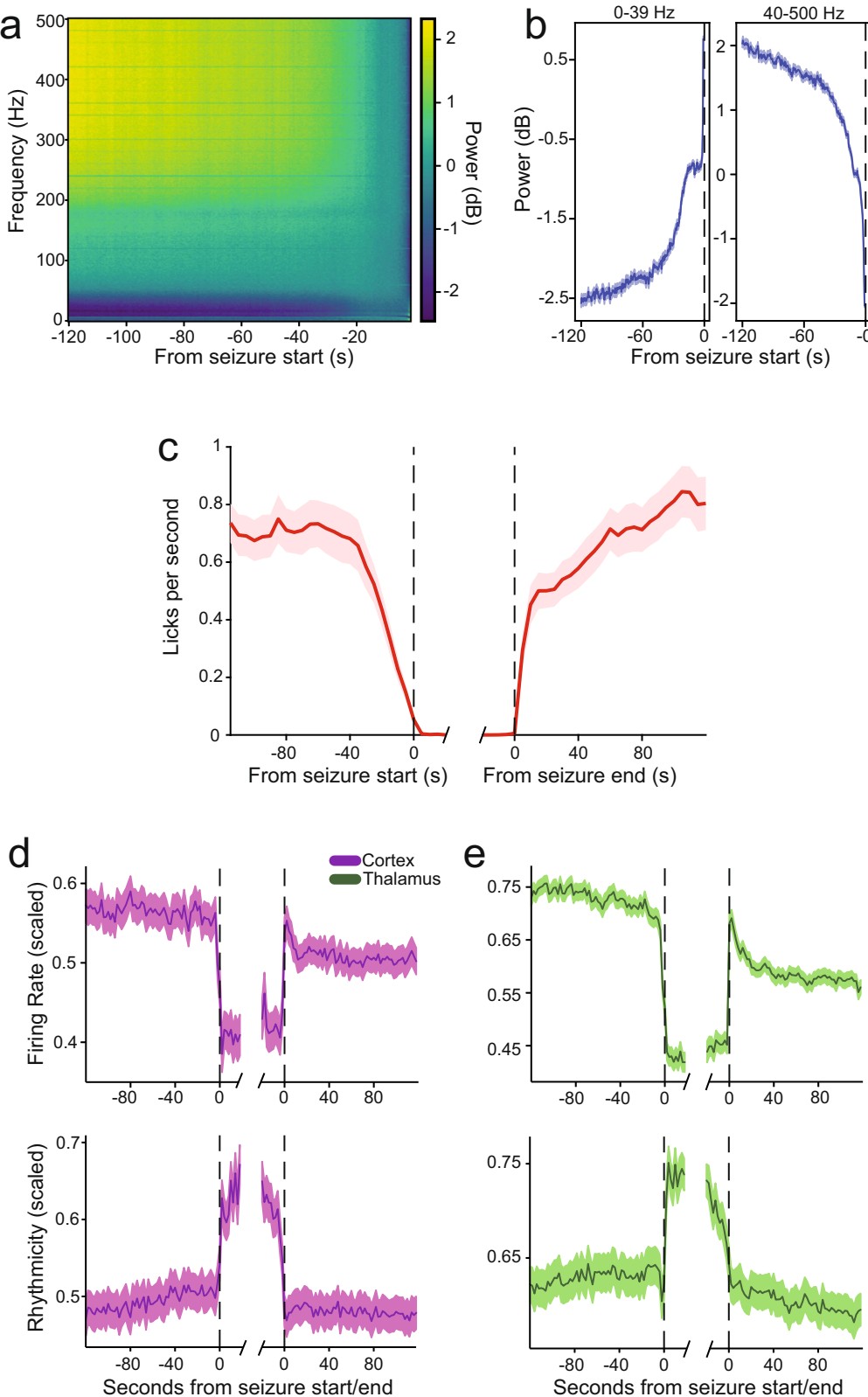

undrugged state resembling human absence seizures including cortical fMRI decreases and thalamic increases[14,20–24]. We then replicated behavioral deficits similar to the interruption of responsiveness and impaired directed activity seen in human absence epilepsy[14,15,19,20]. This was accomplished by reducing auditory stimulus intensity and increasing interstimulus intervals to minimize seizure interruption, revealing impaired responses on two tasks during seizures. Having

established these similarities to studies of impaired consciousness in human absence epilepsy, we investigated the neuronal basis of impaired brain function during SWDs. We found, using ensemble unit activity recordings, that overall cortical and thalamic neuronal firing is decreased during seizures. This broadly decreased activity differs from common concepts of seizures as paroxysmal events involving abnormally increased brain activity. However, it should be noted that

**Fig. 6 | Electrophysiological and behavioral changes tens of seconds before seizure initiation. a** Mean spectrogram of frontoparietal electroencephalogram (EEG) over 120 s prior to seizure initiation in the sustained motivated licking paradigm. Warm colors indicate higher values and cool colors lower values. Mean power (dB) in the 120 s to 1 s time period prior to each seizure was divided by mean power over the same time period for each frequency in the spectrogram, and log tranformed ($10 \times \log_{10}$) as described in the Methods. The resulting spectrograms were averaged across seizures within each animal (26,935 seizures total) and then across animals ($n = 27$ animals). **b** Time courses of power in low (0–39 Hz) and high (40–500 Hz) frequency bands before seizure, using the same data as **a**. **c** Mean lick rate over 120 s prior to seizure initiation in sustained motivated licking paradigm ($n = 27$ animals, 26,935 seizures). **d** Somatosensory cortical (purple colors) firing rate and rhythmicity index (in 2 s bins) in the 120 s around seizure initiation and termination ($n = 165$ cells, 21,866 seizures), showing gradual decrease in firing rate from −80 to −10s before seizure onset ($p = 0.005$), and gradual increase over the same period in cortical rhythmicity ($p = 1.87 \times 10^{-6}$). **e** Ventrobasal thalamic (green colors) firing rate and rhythmicity index (in 2 s bins) in the same time periods ($n = 163$ cells, 10,676 seizures) showing a gradual decrease in firing rate from −80 to −10s before seizure onset ($p = 9.2 \times 10^{-8}$) but no change over this period in rhythmicity ($p = 0.56$). Values are expressed as mean ± standard error.

decreased total neuronal activity has previously been observed during SWDs in GAERS[10] as well as in the *stargazer* mouse model of absence epilepsy[11].

We also discovered four distinct patterns of neuronal activity changes in both cortex and thalamus during seizures. All neuron types showed increased rhythmicity of firing during seizures. However, we observed that individual cortical and thalamic neurons expressed four different patterns of largely consistent behavior from seizure to seizure—either a sustained decrease, sustained increase, onset peak or no change in firing (SD, SI, OP, or NC neurons). The largest group of neurons in both cortex and thalamus showed the SD pattern, explaining the overall decrease in neuronal firing during SWDs in both regions. In further examining the relative changes in firing during the EEG spike and wave phases of SWDs, we found that the decreased overall firing in SD neurons could be explained by relatively greater decreases in firing during the waves compared to relatively smaller increases during spikes (Fig. 4g, j; Supplementary Fig. 4E, F). Conversely, the increased overall firing in SI neurons was explained by greater firing during spikes relative to decreases during waves. However, the SD neurons were far more numerous than SI neurons, explaining the overall sustained decrease in cortical and thalamic activity during seizures. Our results add to the established characteristics of GAERS cortical neurons, including hyperexcitability and changes in synchrony initiated in a focal region but with consequences on broad networks[8,52]. The present findings suggest that the dominant effects of SWD on neuronal circuits, disrupting functions including consciousness, are enhanced rhythmicity and synchrony affecting all neuronal types, as well as overall decreased mean neuronal activity arising primarily from SD neurons.

Further work is needed to histologically and functionally identify these distinct neuronal types, which was not possible with the present silicon probe multi-contact recordings. Relevant approaches may include juxtacellular recordings with labeling, and simultaneous recording of paired neurons to assign excitatory and inhibitory identities[53,54]. Our recordings also revealed additional intriguing findings at the time of SWD initiation. We observed a rapid dip in firing 2–3 s before SWD initiation, followed by a transient peak in firing coinciding with SWD onset (Fig. 4a, c). The antecedent dip in firing was seen in nearly all neuronal types, however interestingly the initial peak was seen only in OP cortical and thalamic neurons. Specifically, we found that transiently increased firing in OP neurons was due to both a higher oscillation frequency and more robust neuronal firing during EEG spikes in the first second of the SWD (Fig. 5c). The unique features of neuronal firing dynamics around the time of SWD initiation can be described with a "ski jump" metaphor, consisting of a broad decrease just before onset, followed by a transient upward deflection and then finally a long downward decline.

These four neuronal groups appear to be differentially responsible for the primary neural characteristics of SWDs in both cortex and thalamus. Specifically, the SD group explains the overall decrease in firing, the OP group explains a peak in the first second of SWDs, and all groups contribute to the dip immediately preceding SWD and to the increase in rhythmicity of firing during SWD. These different neuronal groups may serve as specific targets for future therapies aimed at modulating different pathophysiological aspects of SWDs[55,56]. Excitingly, suppression of rodent SWDs with cell-type specific interventions has been demonstrated[57,58], and our results may provide an avenue to establishing the behavioral translatability of such interventions. In this case the populations we have observed must first be characterized before they may be targeted for selective modulation. The first second of the oscillation, and the OP group that distinguishes it, may be an attractive initial focus: if these neurons can be prevented from peaking in firing, it may affect the probability of seizure onset and progression. In addition, if a specific neuronal subtype could be associated with the state changes preceding seizures, or with the impairment of consciousness, those neurons could be specifically targeted to prevent these pathological features. While diversity of cortical activity in absence seizures is being increasingly recognized[11,59], distinct functional groups have not been previously characterized.

An even more attractive focus of therapy may be the prevention of seizures from starting in the first place, and this study demonstrates electrophysiological and behavioral changes tens of seconds—-1 min—before seizure initiation. We observed that 40–60 s before obvious SWD onset, there was a progressive decrease in overall neuronal firing rate, broadband gamma (>40 Hz) power on EEG, and spontaneous motivated licking behavior; accompanied by an increase in neuronal firing rhythmicity and in low frequency EEG power (Fig. 6). These findings combined suggest a prolonged state change preceding seizure initiation, perhaps involving decreased arousal, implying that these changes may make brain networks particularly vulnerable to absence seizures. Indeed, previous work in human patients and absence rodent models suggests that seizures are more common in awake states with decreased arousal or reduced attentional demand[18,19,28,40,41,50]. Prior work has shown that EEG changes can precede the obvious onset of SWD by a few seconds[60–64], but has not examined the much longer-term progressive changes observed here. Human neuroimaging and hemodynamic studies have shown changes up to 10 s preceding SWD onset[14,20,42,43]. Most human behavioral studies show abrupt onset of deficits confined to the ictal period of absence seizures[14,15,65–67], although some older studies showed more subtle impairments of behavior, which may precede SWDs by a few seconds[68–70]. The longer-term changes in electrophysiology and behavior observed in the present work should be investigated further in both human patients and animal models to better understand the pathophysiological state of neural networks which predispose to seizure initiation, and potential prevention.

The behavioral impairments that we observed in GAERS during SWDs provide an important link to studies of impaired consciousness in human absence seizures. Human investigations of impaired consciousness in absence seizures have used a variety of tasks to demonstrate impaired responses to external stimuli or an arrest of ongoing interactive behaviors with the environment[13–18]. However, previous studies in absence epilepsy rodent models did not succeed in replicating these findings, perhaps because seizures are often suppressed during active engagement in a task, making testing in rodent models challenging[28,40,41]. We found that by reducing auditory stimulus intensity and increasing interstimulus intervals we avoided overly arousing the animals, so that behavior during spontaneous SWDs

could be investigated effectively. Similar to impairments seen in human patients, GAERS exhibited clear deficits during SWDs in two tasks, one showing decreased responses to auditory stimuli, and another showing deficits in ongoing spontaneous motivated licking behavior (Fig. 2). Interestingly, in human absence seizures, behavioral responses can sometimes be relatively spared, varying from spared to impaired from seizure-to-seizure even in the same individual (reviewed in ref. 18). Human studies have demonstrated that impaired responses are associated with more physiologically severe SWDs, for example showing greater EEG power in the frequency range of spikes and waves, compared to SWDs with spared behavioral responses[14–16,19]. In the GAERS model we similarly found that some SWDs showed spared behavioral responses, and that these SWDs had reduced EEG power in the frequency range of spikes and waves (Fig. 2g, h), further strengthening the resemblance to human absence epilepsy. An important direction for future work will be to investigate the relationship between variable behavioral impairment and neuronal subpopulation activity during seizures.

The main neuronal activity changes we observed in both the cortex and thalamus during absence seizures in the rodent model were overall reduced firing and increased rhythmicity. Numerous previous studies have shown increased corticothalamic rhythmicity in animal models of absence epilepsy[7,10,12,71–75]; but relatively few have examined the relative overall level of activity in these circuits, to determine if it is pathologically increased or decreased during seizures compared to baseline. Seizures are commonly thought to represent pathologically increased neuronal firing, although it is recognized that inhibition can occur as well during seizures even in their fundamental definition[76]. Interestingly, a recent study in the *stargazer* mouse absence model using GcaMP imaging in the visual cortex demonstrated reduced activity in most neurons during seizures[11]. This is in general agreement with our measurements in the cortex of GAERS and with human fMRI studies showing predominantly decreased activity in the cortex during absence seizures[14,20–24] including studies that related these changes to impaired consciousness[14,19,20]. Our neuronal recordings and fMRI measurements done in GAERS in the awake state are in agreement in the cortex, both showing overall reduced neuronal activity during absence seizures. We suggest that the contrast with previous animal model SWD hemodynamics is due to the lack of anesthetic or sedative agents in our preparation, and consequently the different baseline arousal states, levels of neural activity, and neurovascular demand.

However, in the thalamus of GAERS in the awake state our neuronal recordings show reduced neuronal firing, whereas our fMRI measurements show signal increases. Interestingly, human fMRI during absence seizures also show prominent thalamic signal increases, although the neuronal basis for these signal increases is not known, and could either represent neuronal excitation or robust inhibitory activity[14,19–24]. It is an important finding that in the awake GAERS model, thalamic fMRI increases are associated with neuronal firing decreases. Previous work has shown that, especially in pathological states such as epilepsy, paradoxical uncoupling or even reverse coupling can be observed between fMRI signals and neuronal activity, which is why direct neuronal recordings are so valuable[36,77–79]. The mechanism for the reduced neuronal firing we observed here in the thalamus in the face of increased fMRI signals is not known. Prior animal model studies of SWDs were done with anesthetic drugs which can abnormally enhance neuronal excitability[35–38,39,44,80], but that was not the case in the current study where animals were undrugged. In fact, prior direct recordings from the thalamus in rodent models with anesthetic agents showed neuronal firing increases during SWDs[36,73]. In contrast, here we found that without anesthetic agents neuronal firing is decreased in the thalamus during SWDs, despite the fMRI increases. The demonstration of this clear disagreement between fMRI and electrophysiology in the animal model raises the possibility that in humans as well, fMRI increases in the thalamus might not represent increased neuronal firing during absence seizures. The thalamus is known to have powerful inhibitory synaptic inputs onto thalamocortical cells arising both from the thalamic reticular nucleus and intrinsic local inhibitory neurons[81]. Because neurovascular coupling in fMRI signals depends on energy consumption due to both local neuronal firing and synaptic activity[82–86], it is possible that the increased fMRI signals in the thalamus during SWD represent massive thalamic inhibitory synaptic inputs[87], despite reduced local neuronal firing. A future investigation of BOLD fMRI and neuronal activity in anesthetized GAERS may, by contrast with our current results, help to elucidate the exact effects of anesthesia on the neural and hemodynamic correlates of SWDs.

In summary, these results demonstrate that the GAERS model reproduces key fMRI and behavioral characteristics of consciousness-impairing human absence seizures, enabling us to elucidate decreased neuronal firing in both cortex and thalamus during seizures as a major pathological factor. The identification of distinct groups of cortical and thalamic neurons with characteristic firing patterns contributing to changes prior to, at the initiation of, and during the sustention of absence seizure discharges may offer long-term therapeutic targets for this disorder. We hope that with further investigation of the neuronal basis of absence seizures, improved treatments can be achieved leading to better patient quality of life.

## Methods

### Animals

Experiments were carried out under approval by the Yale University Office of Animal Research Support. All experiments were performed with Genetic Absence Epilepsy Rats from Strasbourg (GAERS), an established polygenic rat model of absence seizures[88], between 3 and 7 months of age. Female animals were used because their smaller adult body size was conducive to the restricted bore size for MRI experiments. Animals had access to food and water ad libitum unless otherwise noted, and were kept on a 12:12 h light:dark cycle. Animals were group-housed prior to surgery and single-housed thereafter to prevent injury or damage to implants.

### Anesthesia for surgical implants

All implantations of fMRI head-fixation hardware, electroencephalography (EEG) electrodes, and unit recording assemblies were carried out under isoflurane anesthesia. However, all experiments were done in the awake unanesthetized state. Anesthesia was induced at 5% isoflurane in 2 liters/minute oxygen in a plexiglass chamber and maintained via a face mask with scavenging. Isoflurane percentage was gradually reduced to 1.5–2% in 1 liter/minute oxygen for maintenance, continuously checking for appropriate surgical depth via breathing rate and pedal reflex. Analgesia was provided by intraperitoneal carprofen injection at 5 mg/kg every 24 h starting 1 h before surgery and ending 48 h post-surgery.

### fMRI with simultaneous EEG acquisition

Eighteen animals were habituated to body and head restraint by gradual acclimation (Supplementary Fig. 1A–C). This took the form of handling for 2 weeks (increasing from 5 min to ~1 h daily), followed by full body restraint in a cotton towel with Velcro straps for a further 2 weeks. Animals were positioned in the towel by brief isoflurane anesthesia so that their body and limbs could be arranged comfortably. Velcro straps were secured snugly around the hips and shoulders and loosely around the neck. Breathing was continuously monitored and if any signs of difficulty were noted the restraints were removed. Time in restraint was increased from an initial 5 min to a final 90 min in 5–10 min increments over the 2 weeks.

A 3D-printed (Formlabs, Form2) MRI-compatible resin headplate (Supplementary Fig. 1C) was affixed to the skull surface using C&M Metabond dental cement under isoflurane anesthesia (see above) after body restraint habituation was completed. The headplate had 3-point

fixation: 2 plastic screws caudally and a 5 cm hard plastic rod screwed into the most frontal part of the headplate to effectively eliminate pitch, roll and yaw movement artefact. To record EEG, the exposed ends of 1mm-diameter carbon fiber wires (WPI, Sarasota, FL) were attached to the frontal and parietal plates of the skull beneath the skin with tissue glue (3 M Vetbond). Signals were acquired as frontoparietal differential recordings. After 5–7 day recovery from surgery, head-fixation restraint was gradually introduced together with body restraint, progressing from intermittent manual holding of the head-plate to screw-fixed attachment to a custom-designed plastic body holder over the course of 1 week. During this period the rat was also gradually habituated to intensity-matched audio recordings of the MR system.

Recordings were acquired over a 2 week period following the end of habituation. All data were obtained on a 9.4 T Bruker horizontal-bore spectrometer, interfaced with Bruker ParaVision v6.0.1 software running on CentOS (Bruker, Billerica, MA, USA) using a custom-built $^1$H radiofrequency transceiver quadrature coil (2 perpendicular coils with 35 mm diameter each). Magnetic field homogeneity was optimized by localized shimming (STEAM sequence using $B_0$ field map) to yield typical water spectrum line-width of less than 30 Hz for the whole brain. High spatial resolution anatomical images were acquired with 12 interlaced contiguous slices in the coronal plane using a T2-weighted TurboRARE sequence. The field of view was 25.6 × 12.8 mm, with 0.1 × 0.1 mm in-plane resolution and 1 mm slice thickness (echo time, TE = 33 ms, repetition time, TR = 2500 ms, number of echos, NE = 8). The single anatomical volume was acquired within 4 min. 12 identically localized functional image slices were captured by gradient-echo echo-planar imaging (GE-EPI) in the coronal plane with the following parameters: TR = 1 s, TE = 10 ms, in-plane resolution 0.4 × 0.4 mm, slice thickness 1 mm. Individual functional image acquisitions (or "runs") lasted for 5 min (300 samples) or 10 min (600 samples). Between 1 and 4 runs were acquired in each recording session depending on apparent EEG quality and animal comfort in the magnet as indicated by lack of movement, and between 1 and 4 recording sessions were acquired from each animal. EEG was simultaneously acquired from the carbon filament electrodes between 0.1 and 300 Hz (Grass Instruments Model 79D Data Recording System), and digitized via a Cambridge Electronic Design Micro 1401 with Spike2 software at 1000 samples/second. Each recording session lasted up to 1 h, to allow insertion and removal within the 90 min maximum training time (see above). SWDs began to appear after a mean of 25 ± 1.8 min, and there was an average of 31.3 ± 3 SWDs per hour, lasting on average 15.1 ± 1.7 s. The corresponding values for the behavioral recordings are first SWD at 30.3 ± 1.8 min, SWD rate of 60.5 ± 2.8 per hour, and SWD duration of 8.7 ± 0.2 s.

As in our prior work with simultaneous EEG-fMRI in rodent models several measures were employed to reduce artifacts induced in the EEG recordings by high field gradient and radiofrequency pulses in the scanner, including selection of EEG amplifiers to reduce artifact, careful placement, and mechanical buffering of MRI-compatible recording leads, and post-acquisition digital smoothing of EEG signals to remove residual high frequency artifacts[35,36,44,78]. These measures allowed spike-wave-discharges to be readily identified in the EEG and labeled as described below (see "Spike-Wave Discharge Marking and EEG Artifact Rejection") (Supplementary Fig. 1D).

### Behavioral data acquisition with simultaneous EEG and behavioral data analysis

Animals were food restricted to 90% of their initial body mass in order to increase motivation to work for a calorific reward. This 90% target mass was maintained through training, and food restriction ceased ~1 week before implantation to allow recovery for surgery. Animals were individually housed and weighed daily, and any animal exceeding 15% loss of initial mass was returned to ad libitum food access. Upon reaching 90% of initial mass animals were introduced (in their home cage) to 20% sucrose (weight per volume) in water, manually dispensed from a 1 ml syringe (Becton-Dickinson). When rats consumed 2 ml of sucrose water within a 10 min presentation session (training stage 0) they were deemed to have learned the association and proceeded to training in the behavioral paradigm.

Both behavioral paradigms were carried out in custom operant chambers (Med Associates Inc.) (see Fig. 2a, b) during the light phase (7am to 7 pm) of the 12:12 h cycle. Med-PC V Software was used to register lick responses, as well as to control and register the timing of events in the operant chamber including auditory stimuli, session start, session end, and rewards. Rewards consisted of a 90 μL bolus of sucrose water. Behavioral sessions were done once per day lasting up to 2 h, or when the animal received 50 rewards (4.5 mL), whichever occured sooner. Training in the behavioral tasks was performed prior to implantation of the EEG recording electrodes.

For the sensory tone detection task (14 animals), rats were trained to respond to a 0.5 second, 8 kHz tone (Med Associates Programmable Audio Generator, chosen as the frequency of maximum sensitivity) by licking at a port (Med Associates Lickometer) within 10 s of the tone onset in order to receive a reward (see Fig. 2a). Training in the sensory tone detection task proceeded in stages. Training stage 1 consisted of a single session in which the animals were manually presented with a 20% sucrose water-filled syringe and introduced to the operant chamber and the lickometer/reward delivery port. Training stage 2 involved the co-presentation of the tone and a reward at 60 s intervals, with no action by the rat required to trigger reward delivery. In training stage 3 presentation of a reward was contingent on triggering of the lickometer within 10 s of onset of stimulus tone presentation. In training stage 4 the intensity of the auditory stimulus was titrated downwards from an initial 80 to 45 dB, the minimum found to maintain performance levels in pilot animals (based on their ability to reach the 50 reward threshold prior to the 2 h time threshold), and the interstimulus interval was increased to a random range from 150 to 210 s (or the automated detection of a SWD based on EEG amplitude, whichever came sooner). Rats were advanced from stage 2 to 3, stage 3 to 4, and stage 4 to implantation and testing when two consecutive sessions were terminated by reaching the 50 reward threshold prior to 2 h. Training in the sensory tone detection task typically took 2 weeks to complete.

For the sustained motivated licking paradigm (27 animals), rats were trained to lick nearly continuously to obtain an intermittent unpredictable reward bolus, which became available for 10 s periods at random intervals varying from 150 to 210 s (see Fig. 2b). Training stage 1 for this paradigm was the same as for sensory tone detection. In Stage 2 rewards became available, with no tone or other indicator, at 30 s intervals. In Stage 3 inter-reward intervals were increased to 1 min, and in Stage 4 to the final range of intervals described above. As for the sensory tone detection task, rats advanced from one stage to the next and then to implantation and testing when two consecutive sessions were terminated by reaching the 50 reward threshold before 2 h had elapsed.

Following training on either task, animals were implanted with frontal, parietal and cerebellar epidural screw electrodes (0–80 × 3/32, PlasticsOne) and a plastic pedestal connector (PlasticsOne) under isoflurane anesthesia for EEG recording and the identification of SWDs. Frontoparietal differential EEG was collected from the epidural screw electrodes via a Model 1800 Microelectrode AC Amplifier and Headstage (A-M Systems) between 0.1 and 1000 Hz with 1000× gain, using the cerebellar electrode as ground. The signal was then digitized at 1000 samples/second with a Micro1401 acquisition unit (Cambridge Electronic Design) and saved as waveform channels in Spike2 v8 (Cambridge Electronic Design). Transistor-transistor logic (TTL) pulses from behavioral Med-PC V Software (Med Associates, Inc.) were recorded as event channels in the same recordings for synchronization of behavioral and electrophysiological data.

Animals recovered for 5 days following the implant and then underwent behavioral retraining, using the same criterion of reaching 50 rewards within 2 h. In all cases animals achieved this reward threshold in a single session. Behavioral testing for experimental data collection began 1 week post-implantation, with 1 daily session per animal. For the sensory tone detection task, the tone was again delivered at random intervals varying from 150 to 210 s, and also whenever a SWD was detected by a custom-written online amplitude threshold-based algorithm (implemented in Spike2, Cambridge Electronic Design). Briefly, this algorithm required the user to set an amplitude threshold on the online broadband (0.1–100 Hz) filtered frontoparietal EEG, placed above resting baseline EEG amplitude. When the mean RMS amplitude of the signal exceeded this threshold for 0.5 s, a SWD was indicated. While this simple method did produce false positives, these simply resulted in 0–5 extra baseline (non-seizure) stimuli per session, in addition to the baseline stimuli delivered at random 150–210 s intervals. We found this method delivered a sufficient number of stimuli during seizure and non-seizure periods. A lick detected within the 10 s reward availability period and in the same behavioral state (non-seizure or seizure) was classed as a successful response. An absence of licks within the 10 s period was classed as an unsuccessful response. A lick detected within the 10 s but after a transition to a different behavioral state (seizure to non-seizure, or non-seizure to seizure) was considered ambiguous and so excluded from analysis.

For the sustained motivated licking paradigm, the unheralded reward bolus again became available for 10 s periods at random intervals varying from 150 to 210 s. A seizure was designated "spared" if it included at least 1 lick in the 10 s prior to seizure initiation and at least 1 lick during seizure, "impaired" if it included at least 1 lick in the 10 s prior to seizure and no licks during seizure, and ineligible for analysis if it included 0 licks in the pre-seizure period.

### Neuronal activity data acquisition

Cortical data were collected using movable, microdrive-mounted multi-electrode silicon probes (4 shanks per probe, 8 electrode contacts per shank, yielding 32 contacts total per animal) (Neuro-Nexus) in the GAERS somatosensory cortex (coordinates from bregma AP −3 mm, ML ±3 mm, DV starting at −0.75 mm and ending between −1.5 and −2 mm) and the OpenEphys digitization/acquisition system. Data were acquired from 7 animals over 67 recording sessions, yielding a total of 165 sortable single units (23 ± 19 units/animal, mean ± SD). Signals from each channel were band-pass filtered between 1.1 and 7603.8 Hz and digitized at 30 kHz and 192× gain. Thalamic data were acquired as described in[10] and reanalyzed for the current study. Briefly, the same silicon probes were lowered into the ventrobasal thalamus (AP −3.35 mm, ML 2.9 mm) and data were acquired via a head-mounted Plexon HST/32V-G20 VLSI-based pre-amplifier and digitized via a Plexon data acquisition system at 20 kHz. Thalamic data were acquired from 7 animals over 58 recording sessions, yielding a total of 163 sortable single units (27 ± 29 units/animal, mean ± SD). For both cortical and thalamic experiments data were collected during 2–4 h sessions during which rats were free to explore, rest, and seize in the recording chamber, and during which electrodes could be moved via the microdrive if no neurons were visible online.

In the animals used for cortical neuronal activity recordings, frontoparietal EEG could not be collected by paired screw electrodes due to space constraints on the skull. Therefore, EEG was obtained by the differential of the voltage from a frontal epidural screw implanted as already described (see section on Behavioral Data Acquisition with Simultaneous EEG) and a single recording site on the silicon probe, acquired with the OpenEphys system as described above. For each animal the first silicon probe channel with a functional signal was chosen (usually channel 1). The broadband frontal epidural

screw−silicon probe site signal was down-sampled at the analysis stage to 1000 Hz. In the animals for thalamic neuronal activity recordings, EEG was recorded using frontoparietal screw electrodes.

### Histology

After acquisition of neuronal activity animals were euthanized by sodium pentobarbital overdose and perfused with paraformaldehyde. Brains were sectioned on a vibratome at 25 micron thickness and visually inspected to confirm the placement of the silicon probe in the somatosensory cortex. For thalamic recordings (see ref. [10]) silicon probes were immersed in DiI dye before implantation due to the greater uncertainty of targeting the deeper structure, and sections were cut at 100 micron thickness.

### Spike-wave discharge marking and EEG artifact rejection

Prior to analysis of fMRI, behavior, EEG, and neuronal activity recordings, spike-wave discharges (SWDs) were identified in Spike2[10]. In all cases EEG at 1000 Hz was used for this step (acquired at 1000 Hz for fMRI and behavior, down-sampled by averaging for neuronal activity). Briefly, smooth (moving average where voltage at time t is set to mean of voltages from time t-10 ms to time t+10 ms) and DC removal (subtraction of moving average where voltage at time t is set to original voltage minus mean of voltages from time t-0.1 s to time t+0.11 s) functions were used to reversibly visually clean the frontoparietal differential EEG. Then, a negative amplitude threshold (mean voltage minus 5–7 s.d. of baseline non-SWD EEG) was used to detect putative spike-wave crossing points, defined as whenever the signal exceeded the amplitude threshold moving away from the zero point. The crossing points were then grouped into events based on the intervals between them (maximum time between initial two crossings 0.2 s, maximum time between any two crossings within an event 0.35 s, minimum of 5 crossings per event) and the defined properties of SWDs (minimum duration 0.5 s, minimum inter-SWD interval 0.5 s merging any SWDs with shorter intervals), and subsequently, using a frequency threshold, these events were classified as SWDs (if >75% of inter-crossing intervals were within the 5–12 Hz range) or other (e.g., noise, sleep). Labeled SWDs were then visually inspected for accuracy of SWD detection, as well as onset and end times. Periods of sleep were identified based on sharp increases in the 1–4 Hz frequency band, and excluded from analysis (note that these periods occurred only during neuronal activity recordings, and not during fMRI or behavioral recordings, presumably due to the more stimulating sensory environment in the latter two protocols). Periods of orofacial movement artifact during fMRI acquisition were excluded as described in the next section.

### fMRI post-acquisition pre-processing

Prior to analysis the following steps were taken to ensure functional signal quality, in this order. Drifts in signal over time were corrected by fitting a third-order polynomial to the time course of each voxel intensity, subtracting this fit from the data, and then adding back the mean over time for each voxel such that the procedure did not change the mean. Epochs of noise artifacts generated in the EEG data by orofacial movement during functional data acquisition were manually labeled using the start and end of distinctive high-amplitude ~7 Hz epochs that accompany such movement and excluded from the fMRI data analysis. This step resulted in exclusion of an average of 1.4% of fMRI data from each session. Runs were then reviewed by visual inspection of EPI images (as video) for any additional evidence of movement, which led to exclusion of 20% of fMRI runs (12 of 60).

To analyze data in a common space across animals and across recording sessions, we selected high anatomical resolution images from a single animal and session in our cohort as a template, and coregistered all data to that template. First, rostrocaudal alignment

was performed by visual inspection of each anatomical slice and comparison with both a standard rat brain atlas and the template anatomical image slices to find the best correspondence with the latter. A custom MATLAB script then allowed in-plane rigid body translation of the image by whole voxels in the dorsoventral and mediolateral planes to find the best correspondence with the template. The same translations applied to coregister the anatomical images with the template were then applied to the functional images from the corresponding recording session. Finally, all functional images were spatially smoothed using a Gaussian kernel with dimensions of $1 \times 1 \times 2.5$ mm (mediolateral × dorsoventral × rostrocaudal) full width at half maximum, chosen to correspond to 2.5 × voxel dimensions in each direction.

## fMRI analysis

Analysis of fMRI signals during SWD was done with a general linear model approach using statistical parametric mapping (SPM12; https://www.fil.ion.ucl.ac.uk/spm/software/spm12/) and in-house software with methods similar to those used for fMRI analysis in human absence epilepsy. Data were pre-whitened using an autoregressive model, and a 128 s high-pass temporal filter was applied, both using SPM12. A first level fixed-effects model was used to generate animal-level activation maps by creating seizure regressors for each recording session in an animal and then concatenating them together across sessions. To create the seizure regressors, a boxcar function with onset, offset and duration times corresponding to each SWD as determined from EEG review (see "Spike-Wave Discharge Detection" above) was convolved with SPM's canonical double gamma hemodynamic response function. Marked artifact periods (see above) were then removed from both the model and the data series to preserve temporal alignment. Following estimation of the general linear model for each animal, a one-sample $t$-test was performed for group analysis across animals using a second-order random effects model, with false discovery rate (FDR)-corrected $p$ threshold <0.05 and minimum cluster size of 10 voxels, again similar to methods used in human absence fMRI studies[20,89,90].

To investigate the temporal dynamics of fMRI associated with SWDs without requiring the assumptions inherent in SPM and hemodynamic modeling, BOLD signal was also expressed as percent-change time courses in regions of interest relative to pre-seizure baseline. These regions (somatosensory cortex including all S1 regions from bregma +1 mm to bregma −3.6 mm, ventrobasal thalamus including VPM and VPL from bregma −2.3 mm to bregma −4.16 mm, and medial and lateral caudate/putamen) were drawn on the template brain as polygon masks in MRIcron (Neuroimaging Tools and Resources Collaboratory) based on corresponding sections from a standard anatomical atlas[91]. Values were calculated for each seizure as (signal − baseline)/baseline × 100, where baseline was defined as the 5 s immediately preceding seizure. Group analyses for fMRI time courses (see Fig. 1c) were performed by first pooling results within-animal averaging across seizures, and then statistics were calculated across animals (n = number of animals). Because we observed decreased firing in the ~3 s before seizure initiation (see Fig. 4a, c), we reanalyzed the fMRI time courses with an adjusted 5 s baseline (8 to 3 s pre-seizure) but did not find any apparent differences in the results (Supplementary Fig. 1E).

## Neuronal activity analyses

Action potential spikes were extracted from continuous recordings including both non-seizure and seizure periods and clustered into individual neurons[10] (see Fig. 3b). Briefly, the broadband signal was high-pass filtered and thresholded to extract action potential waveforms, and the first three principal components of each action potential on each channel were used to cluster action potentials, first

via the unsupervised KlustaKwik program (Ken Harris, GNU General Public License) and then via supervised refinement.

Spike trains attributed to individual neurons were labeled as cortical or thalamic based on probe location during acquisition and further analyzed. All analysis steps including deriving firing rates, averaging across seizures, plotting spike-triggered averages, and calculating rhythmicity indices as described below were carried out with custom-written scripts in Python (available on GitHub). Firing rates were calculated as action potentials per second in non-overlapping bins (width 0.5 s for seizure onset and offset analyses, or 2 s for pre-seizure state analyses) averaged across seizures within each neuron. Normalized firing rates were then calculated using the mean firing rate time course for each neuron by dividing the mean firing rate at each time by the maximum firing rate in the epoch being analyzed (including pre-ictal, ictal and postictal time points). Group analyses were then performed by averaging these normalized firing rates across neurons.

To separately analyze the contributions of neuronal activity during the spike and wave phases of the spike-wave complex, we performed spike-triggered averages of neuronal activity centered at the peaks of EEG spikes. The large negative-going spikes of EEG spike-wave complexes were detected with a custom-written algorithm adapted from that previously used[10]. Briefly, after application of smooth and DC remove functions as described above, EEG signals were exported from Spike2 to MATLAB. An amplitude threshold was manually set in the negative direction, and all time points exceeding this threshold within previously-identified SWDs (see Spike-Wave Discharge Marking and EEG Artifact Rejection section above) were marked. All local minima (points of greater negative amplitude than their immediate neighbors) beyond the threshold were identified, and any of those local minima that were within 60 ms of the preceding minimum were discarded to prevent double-detection within a single spike-wave complex. Finally, the global minimum amplitude values within a sliding 100 ms window were marked as the times of the spike-wave complex spikes.

The mean firing rate for each neuron across spike-wave cycles was next calculated in 1 ms bins centered in time around the EEG spike peaks during SWDs, and group analyses were performed by averaging across neurons (see Figs. 4e, h and 5b, c). For comparison we also calculated the firing rate at baseline using all data during non-SWD (and non-artifact) periods, calculating the mean firing rate at intervals with center points spaced at the same duration as the mean interval between SWD EEG spikes, using the same approach as for the SWD spike-centered averages. Overall changes in firing during SWDs compared to baseline may be influenced by both firing rate during the spike phase and the wave phase[71]. Therefore, we calculated relative changes in firing rate during both the spike phase and during the wave phase compared to baseline. Mean relative increase in firing during the spike phase was referred to as "peak surplus" firing, and mean relative decrease in firing during the wave phase was called "trough deficit" firing (see Fig. 4e, h). The peak surplus phase for each location (cortical or thalamic) and category (see "Neuronal Categorization" below) was defined as the period around the EEG spike during which the mean firing for that location/category continuously exceeded baseline. The trough deficit phase was defined as the periods between the EEG spikes (during the EEG waves) during which mean firing was continuously below baseline. See shaded regions in Fig. 4e, h. Peak surplus firing was calculated as the mean total spike count in this period for a given cell minus the mean total spike count in the corresponding period around baseline); similarly, trough deficit firing was calculated as the mean total spike count in the trough periods for a given cell subtracted from the mean total spike count in the corresponding periods around baseline for that cell. Peak surplus and trough deficit firing were both calculated for one full cycle in either direction before and after the central firing peak (see Fig. 4e, h).

To quantify the rhythmicity of action potential firing before, during and after SWD, we defined a "rhythmicity index". This was calculated in 2 s nonoverlapping bins from spike trains by first deriving interspike intervals (differences between spike times), then the coefficient of variation (standard deviation divided by mean) of these interspike intervals, and finally the reciprocal of that coefficient of variation. These were derived for each seizure and then averaged within a neuron and normalized as described for neuronal firing rates, and finally averaged across neurons for group analysis.

### Neuronal categorization

Raster plots and firing rate histograms were constructed across all SWD for each neuron (extended data slides are available at https://doi.org/10.5061/dryad.rfj6q57dz), and neurons were sorted into the following categories based on visual inspection of their firing pattern around seizure initiation: Sustained decrease (SD) neurons showed reduced mean firing rate throughout the SWD relative to pre-seizure baseline; sustained increase (SI) neurons showed increased firing throughout the SWD; onset peak (OP) neurons showed a transient increase in firing within ±1 s of seizure onset (classified as OP regardless of increased, decreased or no change in firing for the remainder of the seizure); no change (NC) neurons showed no change in mean firing rate during SWD compared to baseline. Neurons which did not fit any of these categories remained unclassified (a total of three cortical and one thalamic neurons were unclassified).

### Spectral analyses

Where the power in various frequency bands was of interest, EEG signals were first prepared by user identification and removal of artifact periods (induced by motion or electrical noise) in Spike2. Spectrograms were derived from the signal by the mtspecgramc function in the Chronux toolbox (http://chronux.org/) using the following parameters: time-bandwidth product = 2s-Hz (frequency bandwidth resolution = 2 Hz), tapers = 3, frequency band = 0 to 500 Hz, sampling frequency = 1000 Hz, window size = 1 s, window overlap = 0.5 s. Power in the frequency range of SWD waves (5–9 Hz) and spikes (15–100 Hz) was extracted by averaging over the power values in those frequency ranges across seizures (see Fig. 2g, h). Analyses of long-term state changes in power prior to seizures were performed by first calculating the mean power from 120 to 1 s before initiation of each individual seizure, and then dividing all power values by the mean over the same time range for each corresponding frequency in the spectrogram. The resulting values (power/mean power) were log-transformed ($10 \times \log_{10}$) to obtain power expressed in dB. The log-transformed, baseline-corrected spectrograms were then averaged (mean) across seizures within an animal before averaging across animals. These were visualized using the imagesc function in MATLAB (Fig. 6a). Time course of power in frequency bands of interest (Fig. 6b) was calculated in the same way except that values were averaged over the frequency ranges in question after baseline normalizing and log transformation but before averaging across seizures and across animals.

### Statistical analyses

Statistical analysis of fMRI data with SPM is described above. Otherwise, data are described using mean and standard error of the mean except if explicitly indicated. Group analyses were generally performed by first pooling results across seizures through averaging within-animal or within-neuron depending on the data type. Group statistics were then calculated across animals (n = number of animals) or across neurons (n = number of neurons). Initial pooling across seizures was not done for analysis of EEG power in behaviorally spared and impaired seizures (see Fig. 2g, h), because seizure type was the logical analysis group in that case. Differences between groups were tested by using paired or unpaired parametric (Student's $t$) or non-parametric (Wilcoxon signed rank and

Mann–Whitney U) tests as appropriate for the data in question. Bonferroni corrections were used to adjust $p$-values where multiple comparisons were made, specifically when testing for changes in firing rate, rhythmicity, and spikes per cycle of each cell group (Supplementary Fig. 4). Significant positive or negative trajectories of time courses (Fig. 6) were sought by deriving Pearson's R and associated $p$-value for the time period −80 to −10 s prior to SWD onset, or −40 to 0 s in the case of lick rate.

## Data availability

All raw data used in this study are available at DataDryad.org: https://doi.org/10.5061/dryad.rfj6q57dz. Source Data are provided with this paper.

## Code availability

All codes used in this study are available at https://github.com/BlumenfeldLab/McCafferty-et-al_2022.

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

## Acknowledgements

This work was supported by NIH/NINDS R37NS100901 (H.B.), the Mark Loughridge and Michele Williams Foundation (H.B.), and the Betsy and Jonathan Blattmachr family (H.B.).

## Author contributions

C.Mc.C., H.B., B.F.G., P.H., B.G.S., F.H., V.C., and A.D. designed research and experiments; C.McC., B.G., K.S., R.A., W.I., P.A., E.A.J., P.Vit., J.S., I.G.F., A.Kun., F.D., B.G.S., and P.H. performed experiments; C.McC., B.F.G., R.T., J.J.L., X.Z., P.S., P.Vin., Z.K., J.H.R., A.Khal., K.S., J.S., and F.D. analyzed data; C.Mc.C. and H.B. wrote the manuscript with critical review by all other authors.

## Competing interests
The authors declare no competing interests.
