## [Peer Review File · Nature Communications]

Decreased but diverse activity of cortical and thalamic neurons in consciousness-impairing rodent absence seizuresREVIEWER COMMENTS

Reviewer #1 (Remarks to the Author):

The present manuscript by McCafferty et al addresses a number of important unsolved issues pertaining to the neuronal basis behind the pathophysiology of absence epilepsy including reduced awareness and engaged behavior, as well as the translational value of rodent model organisms that have been used to study this disease for decades. Despite a broad variety of species, recording techniques and analysis methods that many different laboratories have used to understand how absence seizures arise, the authors rightfully point out that there are many inconsistencies and gaps in the literature that need to be overcome in order to bolster the validity of these models for the human disease, and to ultimately fully understand the cellular underpinnings in the rodent models to develop better curative approaches than the often inadequate ones we have today. Therefore, the work is of high importance, and I see no reason why it will not have a significant impact on the field of absence epilepsy research; from a methodological viewpoint its results will be applicable to other epilepsies, and arguably a number of other brain diseases, as well.

The first main result reported here is that it is indeed plausible and reasonable to study behavioral phenotypes of human epileptic disease in a rodent model by demonstrating behavioral arrest during spike-wave discharges (SWDs) while rats were performing two different discrimination tasks. Furthermore, the authors succeeded in recording brain activity in awake epileptic rats using an fMRI scanner with simultaneous EEG, which represents a major technical achievement. Next the authors performed extracellular multi-site electrophysiology using multi-electrode silicon probes in the somatosensory cortex and thalamus of the epileptic rats, and confirmed their fMRI results which showed an overall decrease in activity, although the thalamic recordings did not match, but they do offer potential explanations for this discrepancy. They performed additional analyses of the single-unit data to uncover four independent subgroups of neurons displaying different activity patterns around SWD events, which is another important finding. Finally, they found that both EEG gamma power and single-unit activity of most neurons gradually decreases, and rhythmicity increases, over longer periods (up to 60 sec) before SWDs emerge.

The demonstration of significant performance reduction in rats during spike-wave seizures using two different tasks that do not disrupt the manifestation of absence seizures is an important achievement and carries high significance for the field of absence epilepsy research; and the awake fMRI-EEG paradigm, as well as the silicon probe/EEG recordings, will be of significant interest to many laboratories studying epilepsy and other neurological disorders in rodents. Likewise, the other findings reported here are novel and significant, particularly for guiding further investigations of the cellular mechanisms underlying seizure emergence and the future development of cell-type targeted therapeutic approaches.

For the overwhelming majority of the study's conclusions, the data presented here do satisfy a high level of scientific rigor. There are a few minor exceptions that I pointed out in my specific comments, one of which should be fixed by adding some statistical confirmation that is already in the text to one of the figures, the other ones can be fixed by adjusting the respective text portions.

The data analysis is of high quality and rigor throughout the manuscript, and the writing is generally clear and follows an appropriate logical flow from one section to the next. Interpretation and conclusions are, for the most part, well formulated and appropriately related to the literature. Again, there are a few exceptions, particularly in the introduction and discussion, where there is room to elaborate on certain points. Furthermore, one of the results is described by the authors as "mechanistic", but in my mind their case for truly having discovered an underlying causal mechanism for their observed phenomenon is not very strong in this instance, so I suggested to change the language here to be slightly more conservative. Some figure panels should be rearranged, enhanced or adjusted in certain ways. However, no additional data or major revisions are required for publication in my opinion, although I do give some optional suggestions in my comments.

The experimental methods chosen to address the study's question are fully appropriate and up to date. There are no major concerns about alternative approaches potentially having greater power

to produce more revelatory insights for the questions asked. However, as mentioned above, the language concerning the claims and conclusions needs to be slightly adjusted. The methods section is appropriate in its scope and detail.

Overall, my recommendation is in favor of publication once my concerns have been satisfactorily addressed. My main concerns center around the language the authors use to frame some of the results and conclusions, which need to be altered in a way that clarifies the limitations of some portions of the data and strikes a more conservative tone in some sections. Any suggestions I made that I do not consider essential to carry out to satisfy the authors' claims and conclusions are marked as optional. I would not categorize any of the adjustments I am requesting as "major", as I do not think additional data must be collected, or reanalyzed in any laborious fashion.

Comments and suggestions for the authors:

General comment: with 14 experimenters and 12 data analyzers listed in your contributions, you may consider making a matrix-style contribution list that gives a little more detail on who exactly was responsible for which parts of the study, which would add to overall transparency.

Specific comments on the manuscript:

Introduction:

Line 66: Please provide references to set up the current understanding of what we do know about impaired consciousness during SWD's. Since consciousness is a contentious concept and notoriously difficult to precisely and objectively define, it would be helpful if you could expand a bit on what your exact definition of consciousness in the context of this study is. How is it different from the collection of more concrete behavioral measurements that you are presenting in your study? What is the current consensus in the field about what aspects of consciousness in rodents can objectively be described and measured? A good review to reference would be [Pennartz et al, *Frontiers in systems neuroscience Review*, 2019], but I am sure there are others.

Line 67: It would be helpful to expand this, e.g. by mentioning pharmacoresistance, which has been implicated in a lack of targeted approaches to manipulate specific cell types.

Line 70: This is a pretty broad statement, and I don't know if it might sound a little bit misleading, since there is pretty solid evidence by now from recordings in multiple species indicating that it depends on the brain area, and on the specific measurement modalities. Based on the absence seizure and conscious awareness literature, is this question really relevant (an overall reduction in firing could be the result of many different combinations of suppression in some areas while a few other areas might be stimulated)? Since SWD's are dynamical events, the temporal evolution of area- or cell- type specific activity changes seems more important than the "overall" behavior, unless you can show that it actually predicts behavioral arrest or loss of consciousness more accurately than any other measure. Given that you go on to look at the distinct behavior of specific subclasses of neurons during SWDs, I think focusing on "overall" activity changes alone doesn't serve you well to set up the background here.

Line 84: Please add some more information on what kind of fMRI changes have been reported and how they relate to the present study.

Line 90/91: Some examples of behavioral arrest in absence mouse models have been shown, e.g. in [Tan et al, *PNAS* 2007], I would add this and other relevant references. I think it is fair to say that impairment in cognitive task engagement has not been studied in absence models, but I would make that distinction clearer. Also, pointing out that you are actually disentangling "spontaneous" from "engaged" behavior would probably strengthen your argument that you are making your model more comparable to human studies.

Lines 104,109: Please add references here. The rest of this paragraph is not as clear as it could be. If your point is that there is no consensus and lots of conflicting evidence (and presumably you say how you will eliminate these problems in the next section), then that should be spelled out more clearly. Specifically, you talk about anesthetic effects in the next paragraph, but it's unclear whether the studies you cite in lines 111 and 114 were all carried out under anesthesia or not. Also, which brain areas did they record from, and could that explain the disparate results? Could differences in the human data also originate from brain area specific changes? Please cite other awake and anesthetized rodent model studies that used other recording modalities to contrast with fMRI.

Line 118: Since you are exclusively using GAERS in your experiments, I would add a few sentences here describing what's known about SWD in GAERS from the existing fMRI/e-phys literature.

Line 148: Please add a little more detail on how rhythmicity and FR changed.

Methods:

Line 297: Has this algorithm been published before? If not, it should at least be briefly explained in the supplementary material, or the code should be made available.

Line 365-7: I wonder if specific details on how periods of sleep and/or movement were omitted from the analysis may have been different in comparable studies and whether this could account for some of the differences in overall increase/decrease in reported activity? (optional)

"Statistical analyses": Since presumably you analyzed multiple neurons per animal (whose activity patterns are most likely not independent, especially within the four SWD participation groups), and the number of neurons was probably not the same in each animal, it would be useful for enhanced clarity and transparency to mention here when and how you used corrections for multiple comparisons.

Results:

Line 548: Please add reference(s).

Line 556: Please add a reference for the ferret study. This would also be a good place to again expand a little bit on whether other animal studies, especially using rodents, have shown these changes (specifically in unanesthetized animals)... I think here you should also remind the reader why you chose the GAERS model and briefly summarize what is known about SWD activity in GAERS. This can be very brief here, but then you should expand on it more in the intro/discussion.

Line 569: It would be helpful to know how much a 1% decrease is in comparison with "normal" activity fluctuations during non-SWD periods. Otherwise it's hard to tell how large this decrease is, even if it's highly significant. The same is true for the thalamic increase.

Line 574: Add a reference so the reader can compare with the level of fMRI signal change in humans.

Lines 628-634: Please add what your threshold for classifying an SWD as having "some persistent licking" was, in order to justify how you chose to split them into "spared behavior SWDs" vs "impaired behavior SWDs". Presumably there is a relatively smooth distribution of SWD duration, amplitude and concomitant behavior expression, right? Or does it change rather abruptly, indicating that you always either have a "complete" or an "incomplete" SWD event? If that is the case, it would also be important to know what the neuronal subpopulations you identified are doing differently, especially in the lead-up to those different kinds of events.

Line 647-648: Again, the differences in SWD involvement between brain areas is an important point, and should be clarified here (as I also mentioned in the introduction): 1) please reference the prior work you are referring to. 2) please be more specific than just saying "cortical involvement... is most prominent in ssctx". How? In which models? What kinds of measurements have shown this under which conditions? 3) Please justify why "To avoid potential extreme values in ...most intensely involved areas" is a valid reason to focus specifically on the areas you did record from. This needs to be made more obvious so the reader can follow your logic here.

Figure 2, G and H: Do the black and green lines have shaded error bands like E does? If so, they are hard to see and should be enhanced if possible.

Line 656: I think it should be part of fig 3 instead of a suppl. fig, even though the figure will get large and busy.

Line 664 - 677: This paragraph is a little cumbersome to follow because it combines the rhythmicity result with the peak/trough way of quantifying the SWD firing rate changes. It is inconvenient having to switch back and forth between fig 3 and suppl fig 2. I would suggest moving the "all" panels from suppl fig 2 to fig 3. Alternatively, the rhythmicity results could move to a separate paragraph, if you want to make it a separate point. But if its purpose is basically to set up the peak/trough quantification (and the point that the trough deficit is greater than the peak surplus), then it's fine. Also, the reader should be reminded of the timescales (or frequency range) on which you compute rhythmicity. You do mention your definition of rhythmicity in line 724, but I think it should be referenced here as well (or instead).

Line 702: delete "or"

This and the following paragraphs: After you define SD, SI, OP, and NC you should consistently use only the abbreviations throughout the rest of the manuscript.

Lines 748-767: For this paragraph I have the following comments/concerns:

- 1) I don't think, based on what most readers would consider an appropriate level of mechanistic insight, that you can claim to have identified a mechanism for the transient increase of the OP group. In my opinion that would require you to show e.g. that OP neurons normally receive strong inhibitory input from some class of neurons, and because those specific interneurons are suppressed during this time frame, OP neurons are hyperactive, but then a rebound excitation from another subpopulation stops this excitation, or something along those lines. The analysis you are showing is relevant in that it explains how your peak/trough calculations give you the characteristic OP behavior, which may hint at potential mechanisms behind that behavior, but it does not reveal the underlying circuit or cellular mechanism itself. Please adjust the language accordingly, or provide to the reader your exact definition of "mechanism" as used in this context.
- 2) I would add appropriate plots depicting the statistics you mention throughout lines 754-762 in figure 4.
- 3) Figure 4 c) at first glance looks like it is still aligned with the groups in b), which is a bit confusing. If possible, make it clearer that these panels do not match the categories above or arrange them differently.
- 4) While it makes sense that the high peak surplus in the first second plays a major role in separating the OP neurons from the other groups, I don't understand how the oscillation frequency plays a role. The EEG waveform presumably changes from the first second to the rest of each SWD in the same way, right? So are the OP neurons more strongly coupled to the EEG spikes than the other groups? Otherwise their oscillation frequency elevation in the first second should not be stronger than the other groups, right? Please explain and make adjustments to the text in this section.

Line 771: Please provide a reference.

Line 783: Please add statistical significance and % values.

Lines 787-795: Unless you do not have the data or you feel that it would be completely outside the scope of the study to run this analysis, it would be very intriguing to see how the different SWD-participating subgroups changed over these long pre-seizure periods. This seems like a very obvious piece of information to add, and it would make a useful additional supplementary figure/panel. (optional)

Discussion:

Lines 799-802: I would suggest briefly relating your overall findings to previous studies that have looked at similar questions in humans and rodents, either here or in the following paragraph.

Line 808: Please add a reference.

Line 809: Please elaborate: "those seen in human absence epilepsy", such as....? [Refs]

Line 816: Please add a reference. Also, please mention other previous studies that have shown neuronal suppression during SWDs. While it is interesting to juxtapose this suppression with the opposite phenomenon of hyperactivity in most other epileptic seizure types, I don't think you should omit previous results here that have reported suppression during SWDs, whether in GAERS or other rodent model organisms.

Lines 823-832: Please add more information on how your results compare to previous studies looking at single unit activity in GAERS cortex, such as [Polack et al, J Neurosci, 2007] , [Jarre et al, Cerebral cortex 2017]

Line 834: I agree that this would be an important next step, but isn't it possible to at least separate pyramidal from GABAergic units based on extracellular waveforms? Unless you feel that this is completely outside the scope of this study, this would be an important component to add, even just as a supplementary figure. (optional) If not, this possibility or how to employ other suitable methods should be discussed in more detail.

Paragraph starting at line 860: Please discuss how your observations and proposed future directions might fit with recent studies showing that modulation of neuronal subtypes, such as certain interneurons, can suppress SWDs in rodent models, e.g. in [Panthi & Leitch, Front cell neurosci 2021] or [Clemente-Perez et al, Cell Rep 2017]. Moreover, as mentioned above this section would benefit from an analysis of the four subgroups in terms of their firing patterns during the 60 sec pre-SWD periods. Does the decrease in gamma EEG power point towards a specific class of neurons that are hypoactive? Another point to expand a bit here would be what the roles of the four subgroups of SWD -modulated neurons in the suppression of conscious experience/behavior might be. Do any of these groups coincide more strongly with the behavioral arrest you measured?

Line 922: Please add a reference.

Paragraph starting at line 931: I think [Polack et al, Cerebral cortex 2009] also recorded single unit activity from different thalamic nuclei in anesthetized GAERS, you might want to reference this study for comparison. I suppose you could easily repeat your experiments with anesthesia to test your hypothesis that being awake may be a reason for the dissociation between fMRI and e-phys measurements. (optional). Presumably this is outside the scope of the current study, but it should be discussed.

Reviewer #2 (Remarks to the Author):

This submission by McCafferty et al. is exceptionally significant to the fields of neurobiology of disease, neuroscience of consciousness, and thalamocortical network physiology. Their paper addresses two key problems concerning the understanding of the neurophysiology of absence seizures: 1) the apparent discrepancies between animal absence seizure models regarding cortical activation and consciousness impairment and 2) a mechanistic relationship between cortical and thalamic individual neuronal firing to seizure development and evolution. Importantly, the investigators then successfully link their single neuron activity results to the observed consciousness impairment and cortical activation.

Their experimental methodology is state-of-the-art, sound, and appropriate for the questions asked. In contrast to previous studies, they developed methods for performing the behavioral tests without overly alerting the subjects and accomplished the fMRI and electrophysiological studies without anesthesia, techniques that are crucial for the study of absence seizures and that will undoubtedly be used by others. They applied the expected analyses of their fMRI, behavior, and electrophysiological data, used appropriate statistical tests, and formed conclusions justified by the data. The methods are well-written and provide enough detail to be reproduced (except for some minor suggestions outlined below).

Minor points:

1. Methods: They stated they recorded from the somatosensory cortex representing the trunk (S1TR) rather than the typical perioral region "to avoid extreme values in facial area." What is meant by this? Were they not able to isolate single units from the perioral region?
2. Methods/Results: What cortical layers (supragranular, granular, infragranular) were recorded? It would be helpful to have an example of the histology for both cortex and thalamus in results (possibly in supplemental data).
3. Methods The diagram of the frame Supplementary 1 is difficult to understand – consider adding a picture of the mouse to the frame.
4. Methods Their use of female rats is justified for experimental reasons. I believe the journal requires the abstract to state the sex of the subjects when animals of only one sex were used.
5. Results – can their spike sorting differentiate interneurons from principal neurons? If so, do they cluster within their defined neuronal types?
6. Results – can they determine the timing of spiking in the cortex vs. thalamus – e.g. does cortex precede thalamus as was observed in other studies?
7. Results Figure 3B Why are the spikes separated by PC7 & PC10 rather than PC1-3?
8. Results Figure 3C shows broad band recordings – presumably to highlight the transition from pre-ictal to ictal EEG. However, broad band does not demonstrate the unit recordings. Possibly an inset with the high frequency recordings would be helpful.
9. Typographical errors:
 - a. Fig 3D legend "2-3" (omitted word seconds)
 - b. Line 828 more numerous than SI neurons (should be "than" and not "that")

Reviewer #3 (Remarks to the Author):

The manuscript by McCafferty et al. explore the neuronal basis of impaired consciousness in absence seizures. They used a well-accepted spontaneous rat model for human absence epilepsy and combined EEG recordings with behavioural measures and fMRI of awake, restrained-habituated GAERS rats. The manuscript is well-written. Methods seem well-established and are described in detail. Overall conclusions are sound. I would recommend publication of this paper. The authors convincingly present behavioural loss of consciousness, SWDs on EEG and decreased cortical fMRI signal in agreement with findings in human absence epilepsy. On neuronal level, they identified decreased overall cortical and thalamic firing and increased rhythmicity. Neuronal sorting revealed four distinct patterns of neuronal activity changes, with the biggest subgroup showing sustained decrease of firing.

Major comments

Further characterization of the identified neuronal subpopulations (histology and function) has not been performed in this study. This finding is indeed interesting and warrants further investigations. For now, speculating about using these different neuronal groups as targets for future therapies seems a bit far-reaching, without any further knowledge about respective cell specifics (end of Intro and discussion line 851ff). Based on their experience in the field, could the authors suggest any potential candidates?

The authors established fMRI in awake, restrain-habituated animals in their study. They describe the habituation procedure in detail. The authors are obviously aware of and state that seizures are more likely to occur in a state of relaxed wakefulness. To overcome the issue that SWD may be interrupted due to loud noise or too stressful/demanding stimuli they have developed a closed-loop test of auditory responsiveness by decreasing stimulus intensity and increasing inter-stimulus interval durations for the behavioural part of the study. I am wondering how stress and a higher state of arousal due to restraint and to the unavoidable loud noise during MR scanning has affected the occurrence and duration of SWDs in their experiments. Also, Isoflurane used for positioning of the animals will temporarily suppress SWDs. Specifically, more detailed information regarding when first SWDs are usually observed during a MRI examination, how many SWDs are observed and the mean duration of SWDs in comparison to the behavioral cohort are of interest. This information can not fully be retrieved from the 90 sec exemplary simultaneous EEG recording (in suppl fig 1) nor from the notion that 670 SWDs were observed in 18 animals (in fig 1). Could the authors identify potential differences in baseline arousal state between MRI and behavioural cohort or exclude such by comparing baseline spectrograms?

For behavioural training animals were food restricted. The authors state that they used the initial body mass as reference value. Female rats between 3 and 7 month of age are not fully matured and under a normal ad libitum diet will gain body weight. I guess behavioral experiments lasted in total weeks (from first training session to termination). From an animal welfare perspective, it would be more reasonable to compare body weights to not food-restricted, age-matched animals. To investigate temporal dynamics of fMRI associated with SWD the authors set the pre-seizure baseline = 5 seconds immediately before seizure. EEG recordings revealed a rapid dip in neuronal firing 2-3 sec before SWD initiation (figure 3D und E). How would shifting the onset/pre-seizure baseline affect the outcome of GLM and time course analysis?

Similarly, did the authors try to use the time point of the steep power shift in the high and low frequency band 40-60 sec before SWD onset as a regressor for GLM or take it into account for definition of the pre-seizure baseline in the time course plots?

Minor comments

Methods, line 164 ff - Please provide information about analgesia

Results line 702 - remove "or" after increase

Yale University
School of Medicine

HAL BLUMENFELD, MD, PHD

Mark Loughridge and Michele Williams Professor
Director, Yale Clinical Neuroscience Imaging Center
Professor of Neurology, Neuroscience, Neurosurgery
Epilepsy Service
Yale University School of Medicine
New Haven, CT 06520-8018
Tel: (203) 785-3865
Fax:: (203) 737-2538
E-mail: hal.blumenfeld@yale.edu

RE: NCOMMS-22-25827A

Sept 22, 2022

Dear Reviewers,

Thank you for your careful consideration of our manuscript and your valuable feedback. We have improved the manuscript based on your suggestions. Please see the below response to each of your points, and the associated changes in the revised manuscript.

Reviewer #1:

REVIEWER COMMENT:

The present manuscript by McCafferty et al addresses a number of important unsolved issues pertaining to the neuronal basis behind the pathophysiology of absence epilepsy including reduced awareness and engaged behavior, as well as the translational value of rodent model organisms that have been used to study this disease for decades. Despite a broad variety of species, recording techniques and analysis methods that many different laboratories have used to understand how absence seizures arise, the authors rightfully point out that there are many inconsistencies and gaps in the literature that need to be overcome in order to bolster the validity of these models for the human disease, and to ultimately fully understand the cellular underpinnings in the rodent models to develop better curative approaches than the often inadequate ones we have today. Therefore, the work is of high importance, and I see no reason why it will not have a significant impact on the field of absence epilepsy research; from a methodological viewpoint its results will be applicable to other epilepsies, and arguably a number of other brain diseases, as well.

The first main result reported here is that it is indeed plausible and reasonable to study behavioral phenotypes of human epileptic disease in a rodent model by demonstrating behavioral arrest during spike-wave discharges (SWDs) while rats were performing two different discrimination tasks. Furthermore, the authors succeeded in recording brain activity in awake epileptic rats using an fMRI scanner with simultaneous EEG, which represents a major technical achievement. Next the authors performed extracellular multi-site electrophysiology using multi-electrode silicon probes in the somatosensory cortex and thalamus of the epileptic rats, and confirmed their

fMRI results which showed an overall decrease in activity, although the thalamic recordings did not match, but they do offer potential explanations for this discrepancy. They performed additional analyses of the single-unit data to uncover four independent subgroups of neurons displaying different activity patterns around SWD events, which is another important finding. Finally, they found that both EEG gamma power and single-unit activity of most neurons gradually decreases, and rhythmicity increases, over longer periods (up to 60 sec) before SWDs emerge.

The demonstration of significant performance reduction in rats during spike-wave seizures using two different tasks that do not disrupt the manifestation of absence seizures is an important achievement and carries high significance for the field of absence epilepsy research; and the awake fMRI-EEG paradigm, as well as the silicon probe/EEG recordings, will be of significant interest to many laboratories studying epilepsy and other neurological disorders in rodents. Likewise, the other findings reported here are novel and significant, particularly for guiding further investigations of the cellular mechanisms underlying seizure emergence and the future development of cell-type targeted therapeutic approaches.

For the overwhelming majority of the study's conclusions, the data presented here do satisfy a high level of scientific rigor. There are a few minor exceptions that I pointed out in my specific comments, one of which should be fixed by adding some statistical confirmation that is already in the text to one of the figures, the other ones can be fixed by adjusting the respective text portions.

The data analysis is of high quality and rigor throughout the manuscript, and the writing is generally clear and follows an appropriate logical flow from one section to the next.

Interpretation and conclusions are, for the most part, well formulated and appropriately related to the literature. Again, there are a few exceptions, particularly in the introduction and discussion, where there is room to elaborate on certain points. Furthermore, one of the results is described by the authors as "mechanistic", but in my mind their case for truly having discovered an underlying causal mechanism for their observed phenomenon is not very strong in this instance, so I suggested to change the language here to be slightly more conservative. Some figure panels should be rearranged, enhanced or adjusted in certain ways. However, no additional data or major revisions are required for publication in my opinion, although I do give some optional suggestions in my comments.

The experimental methods chosen to address the study's question are fully appropriate and up to date. There are no major concerns about alternative approaches potentially having greater power to produce more revelatory insights for the questions asked. However, as mentioned above, the language concerning the claims and conclusions needs to be slightly adjusted. The methods section is appropriate in its scope and detail.

Overall, my recommendation is in favor of publication once my concerns have been satisfactorily addressed. My main concerns center around the language the authors use to frame some of the results and conclusions, which need to be altered in a way that clarifies the limitations of some portions of the data and strikes a more conservative tone in some sections. Any suggestions I made that I do not consider essential to carry out to satisfy the authors' claims and conclusions are marked as optional. I would not categorize any of the adjustments I am requesting as "major", as I do not think additional data must be collected, or reanalyzed in any laborious fashion.

AUTHOR RESPONSE:

We thank the reviewer for their careful reading of our manuscript, and for these positive and detailed comments, which are invaluable feedback for the improvement of the manuscript. Detailed responses to the specific comments are below.

REVIEWER COMMENT:

General comment: with 14 experimenters and 12 data analyzers listed in your contributions, you may consider making a matrix-style contribution list that gives a little more detail on who exactly was responsible for which parts of the study, which would add to overall transparency.

AUTHOR RESPONSE:

Thank you for this useful suggestion; a matrix has been added on Pg2.

REVIEWER COMMENT:

Line 66: Please provide references to set up the current understanding of what we do know about impaired consciousness during SWD's. Since consciousness is a contentious concept and notoriously difficult to precisely and objectively define, it would be helpful if you could expand a bit on what your exact definition of consciousness in the context of this study is. How is it different from the collection of more concrete behavioral measurements that you are presenting in your study? What is the current consensus in the field about what aspects of consciousness in rodents can objectively be described and measured? A good review to reference would be [Pennartz et al, Frontiers in systems neuroscience Review, 2019], but I am sure there are others.

AUTHOR RESPONSE:

The following text has been added to the Introduction from line 83 to address this query: "Evaluating consciousness in human and animal model epilepsy research in particular typically involves testing responsiveness and/or ability to report experiences (Giacino et al., 2004), as particular indicators of consciousness (Pennartz et al., 2019) that have direct relevance for the impact of an epileptic seizure on an individual's safety, wellbeing, and ability to interact with their environment. In this study we therefore consider impaired consciousness in this context, focusing on tests of responsiveness more easily assessed in animal models."

REVIEWER COMMENT:

Line 67: It would be helpful to expand this, e.g. by mentioning pharmacoresistance, which has been implicated in a lack of targeted approaches to manipulate specific cell types.

AUTHOR RESPONSE: the following text has been added from line 59 to address this issue:

"This variability, including in the efficacy of gold-standard pharmacological interventions (Shinnar et al., 2017), suggests the need for targeted therapeutics that would in turn require the identification of molecular or cellular mechanistic targets."

REVIEWER COMMENT:

Line 70: This is a pretty broad statement, and I don't know if it might sound a little bit misleading, since there is pretty solid evidence by now from recordings in multiple species indicating that it depends on the brain area, and on the specific measurement modalities. Based on the absence seizure and conscious awareness literature, is this question really relevant (an overall reduction in firing could be the result of many different combinations of suppression in

some areas while a few other areas might be stimulated)? Since SWD's are dynamical events, the temporal evolution of area- or cell- type specific activity changes seems more important than the "overall" behavior, unless you can show that it actually predicts behavioral arrest or loss of consciousness more accurately than any other measure. Given that you go on to look at the distinct behavior of specific subclasses of neurons during SWDs, I think focusing on "overall" activity changes alone doesn't serve you well to set up the background here.

AUTHOR RESPONSE: The text from line 62-66 has been edited to indicate that the investigation of overall changes in firing is a precursor to the investigation of neuronal subpopulations. It now reads "Therefore, understanding the nature of neuronal activity producing impaired consciousness in absence seizures has important translational value, including changes in the overall level of neuronal firing during SWDs with impaired consciousness and, more specifically, whether absence seizures affect activity in all neurons the same way or whether different neuronal subpopulations may display distinct activity patterns."

REVIEWER COMMENT:

Line 84: Please add some more information on what kind of fMRI changes have been reported and how they relate to the present study.

AUTHOR RESPONSE: the specifics of the fMRI changes reported in people with absence epilepsy and their relevance to the study are now described from line 73 of the Introduction as follows:

"Human research has also successfully related the impaired consciousness in absence seizures to functional magnetic resonance imaging (fMRI), showing that the magnitudes of fMRI changes in both cortex and thalamus are associated with the severity of impaired behavioral responses^{14, 19, 20}. Specifically, the cortex in human fMRI studies of absence seizures shows mainly fMRI decreases whereas the thalamus shows fMRI increases^{14, 20, 21, 22, 23, 24}. Investigating the neuronal basis of these changes in a valid animal model could provide important further mechanistic insights."

REVIEWER COMMENT:

Line 90/91: Some examples of behavioral arrest in absence mouse models have been shown, e.g. in [Tan et al, PNAS 2007], I would add this and other relevant references. I think it is fair to say that impairment in cognitive task engagement has not been studied in absence models, but I would make that distinction clearer. Also, pointing out that you are actually disentangling "spontaneous" from "engaged" behavior would probably strengthen your argument that you are making your model more comparable to human studies.

AUTHOR RESPONSE: We thank the reviewer for pointing out this lack of clarity, this has been addressed by a statement from line 94 clarifying that arrest of spontaneous behavior has been observed in animal models including the study by Tan et al, as follows:

"no prior studies have reproduced these findings in an animal model of absence epilepsy, in part because such studies^{27, 28} have tended to focus on the arrest of spontaneous behavior rather than the impairment of engaged behavior."

REVIEWER COMMENT:

Lines 104,109: Please add references here. The rest of this paragraph is not as clear as it could be. If your point is that there is no consensus and lots of conflicting evidence (and presumably you say how you will eliminate these problems in the next section), then that should be spelled out more clearly. Specifically, you talk about anesthetic effects in the next paragraph, but it's unclear whether the studies you cite in lines 111 and 114 were all carried out under anesthesia or not. Also, which brain areas did they record from, and could that explain the disparate results? Could differences in the human data also originate from brain area specific changes? Please cite other awake and anesthetized rodent model studies that used other recording modalities to contrast with fMRI.

AUTHOR RESPONSE: We apologize for the previous lack of clarity. We have clarified that all animal studies of SWD hemodynamics have been carried out under some form of anesthesia or sedation from the Introduction, from line 115 (“This discrepancy may be because all prior animal fMRI studies were done under some form of anesthesia or sedation, or because of a neuronal or hemodynamic difference between animal SWDs and human absence seizures. Prior work has not yet investigated both fMRI changes and direct neuronal recordings under the same conditions in an awake undrugged animal absence epilepsy model.”). We also removed some text about neurovascular coupling that was not directly relevant to this point and therefore potentially confusing. More detailed discussion on this topic is also provided in two paragraphs in the Discussion, on p. 44 from line 1126 and p. 45 from line 1141.

REVIEWER COMMENT:

Line 118: Since you are exclusively using GAERS in your experiments, I would add a few sentences here describing what's known about SWD in GAERS from the existing fMRI/e-phys literature.

AUTHOR RESPONSE: We have added some detail on previous fMRI and electrophysiological characterization of SWD in GAERS in the Introduction, from line 125 as follows:

“Prior work in GAERS has demonstrated seizure-related rhythmic activity in cortical and thalamic neurons but has not investigated fMRI and neuronal changes without anesthetic drugs^{8, 10, 46}”

REVIEWER COMMENT:

Line 148: Please add a little more detail on how rhythmicity and FR changed.

AUTHOR RESPONSE: We have added the direction of change of rhythmicity (increase) and FR (decrease) during the 40-60s before seizures in the Introduction from line 181, specifically stating “... neuronal rhythmicity increased and firing rates decreased over similar time scales.”

REVIEWER COMMENT:

Line 297: Has this algorithm been published before? If not, it should at least be briefly explained in the supplementary material, or the code should be made available.

AUTHOR RESPONSE:

This algorithm was implemented online in Spike2 only and was tailored to this particular task, as is now explained from line 337: “Briefly, this algorithm required the user to set an amplitude threshold on the online broadband (0.1 – 100 Hz) filtered frontoparietal EEG, placed above resting baseline EEG amplitude. When the mean RMS amplitude of the signal exceeded this threshold for 0.5 seconds, a SWD was indicated. While this simple method did produce false positives, these simply resulted in 0-5 extra baseline (non-seizure) stimuli per session, in addition to the baseline stimuli delivered at random 150 – 210 s intervals.”

REVIEWER COMMENT:

Line 365-7: I wonder if specific details on how periods of sleep and/or movement were omitted from the analysis may have been different in comparable studies and whether this could account for some of the differences in overall increase/decrease in reported activity? (optional)

AUTHOR RESPONSE:

This is an important point, thank you for raising it, and particularly relevant for our fMRI measurements where our results done without anesthetic drugs differ from other studies done with anesthetic drugs. As is now clarified in the Methods from line 415 we did not observe periods of sleep in our fMRI recordings, and we would hypothesize that this constitutes a significant difference from previous hemodynamic recordings in which anaesthetics or sedatives were used (“note that these periods occurred only during neuronal activity recordings, and not during fMRI or behavioral recordings, presumably due to the more stimulating sensory environment in the latter two protocols”). Specifically, the baseline arousal levels and consequently neural activity and neurovascular demand may have been quite different. This is also now discussed in the Discussion section from line 1126 (“We suggest that the contrast with previous animal model SWD hemodynamics is due to the lack of anesthetic or sedative agents in our preparation, and consequently the different baseline arousal states, levels of neural activity, and neurovascular demand.”). Movement artifacts by contrast were removed only during the fMRI recordings and again constitute a possible point of difference from previous recordings, but we consider this unlikely as their anesthetized/sedated preparations would not have had substantial movement in the first place.

REVIEWER COMMENT:

“Statistical analyses”: Since presumably you analyzed multiple neurons per animal (whose activity patterns are most likely not independent, especially within the four SWD participation groups), and the number of neurons was probably not the same in each animal, it would be useful for enhanced clarity and transparency to mention here when and how you used corrections for multiple comparisons.

AUTHOR RESPONSE:

We thank the reviewer for this suggestion, we have employed the Bonferroni correction where multiple comparisons were applied and added details on this in the Methods from line 593: “Bonferroni corrections were used to adjust p-values where multiple comparisons were made, specifically when testing for changes in firing rate, rhythmicity, and spikes per cycle of each cell group (Supplementary Fig. 4).”.

REVIEWER COMMENT:

Line 548: Please add reference(s).

AUTHOR RESPONSE:

Apologies for this and other omitted references due to a mistake with the citation program; these have now been rectified.

REVIEWER COMMENT:

Line 556: Please add a reference for the ferret study. This would also be a good place to again expand a little bit on whether other animal studies, especially using rodents, have shown these changes (specifically in unanesthetized animals)... I think here you should also remind the reader why you chose the GAERS model and briefly summarize what is known about SWD activity in GAERS. This can be very brief here, but then you should expand on it more in the intro/discussion.

AUTHOR RESPONSE: A reference has been added, and justification for the choice of GAERS has been added from line 616: "The GAERS model was chosen due to its exclusive expression of typical absence seizures^{12,49}, the greater tractability of rats (vs. monogenic mouse models) to behavioral interrogation, and the etiological advantage of genetic vs. pharmacological models of absence seizures¹²."

REVIEWER COMMENT:

Line 569: It would be helpful to know how much a 1% decrease is in comparison with "normal" activity fluctuations during non-SWD periods. Otherwise it's hard to tell how large this decrease is, even if it's highly significant. The same is true for the thalamic increase.

AUTHOR RESPONSE:

We thank the reviewer for this suggestion; we have now added comparative values from other studies together (including "normal" non-epileptic changes) together with the missing reference noted below (plus additional references) from line 642: "The magnitude of the changes, at approximately 1%, are comparable to the ~0.5% changes observed in humans^{14,20} and at the upper end of the general 0-1% expected range for normal (non-noise-induced) fMRI signal dynamics across species^{55,56}. The relatively large changes in GAERS may be due to their selective breeding for the expression of robust, widespread SWDs."

REVIEWER COMMENT:

Line 574: Add a reference so the reader can compare with the level of fMRI signal change in humans.

AUTHOR RESPONSE: Please see the above response for the preceding comment.

REVIEWER COMMENT:

Lines 628-634: Please add what your threshold for classifying an SWD as having "some persistent licking" was, in order to justify how you chose to split them into "spared behavior SWDs" vs "impaired behavior SWDs". Presumably there is a relatively smooth distribution of

SWD duration, amplitude and concomitant behavior expression, right? Or does it change rather abruptly, indicating that you always either have a “complete” or an “incomplete” SWD event? If that is the case, it would also be important to know what the neuronal subpopulations you identified are doing differently, especially in the lead-up to those different kinds of events.

AUTHOR RESPONSE:

Thank you for this suggestion; we have added the following clarifying text to the Results section, from line 700: “Behaviorally “spared” seizures included at least 1 lick during the seizure, whereas “impaired” seizures were defined as those with no licks (see Methods, Behavioral Data Acquisition with Simultaneous EEG and Behavioral Data Analysis section for details).” We have also added Supplementary Fig. 2 to show the distributions of SWD duration and voltage amplitude for spared and impaired SWD, now referred to in the Results section on line 708. As our dataset did not include simultaneous single neuron recordings and behaviour we cannot investigate the subpopulations’ relationships to spared and impaired SWDs, but we believe this to be a critical direction for future studies. We now add this to the Discussion, from line 1107 stating “An important direction for future work will be to investigate the relationship between variable behavioral impairment and neuronal subpopulation activity during seizures.”

REVIEWER COMMENT:

Line 647-648: Again, the differences in SWD involvement between brain areas is an important point, and should be clarified here (as I also mentioned in the introduction): 1) please reference the prior work you are referring to. 2) please be more specific than just saying “cortical involvement... is most prominent in ssctx”. How? In which models? What kinds of measurements have shown this under which conditions? 3) Please justify why “To avoid potential extreme values in ...most intensely involved areas” is a valid reason to focus specifically on the areas you did record from. This needs to be made more obvious so the reader can follow your logic here.

AUTHOR RESPONSE:

Thank you for pointing out this lack of detail – we have now clarified exactly how the peri-oral somatosensory cortex differs from other regions in SWDs and why we would therefore want to avoid it from line 721: “Specifically, deep layer peri-oral somatosensory neurons appear to initiate SWDs in GAERS, and this region of cortex temporally leads SWD activity in the related WAG/Rij model. Because our focus was on general cortical activity rather than on focal seizure initiation, we therefore recorded from subgranular layers (0.9 to 2 mm below pial surface) of somatosensory cortex in the trunk region (Supplementary Fig. 3).”.

REVIEWER COMMENT:

Figure 2, G and H: Do the black and green lines have shaded error bands like E does? If so, they are hard to see and should be enhanced if possible.

AUTHOR RESPONSE:

Thank you for this observation, there were indeed shaded error regions in G and H; we have adjusted the scaling of the figure to make them more visible. At some points the error is still very small due to a large sample size and lack of variability.

REVIEWER COMMENTS:

Line 656: I think it should be part of fig 3 instead of a suppl. fig, even though the figure will get large and busy.

Line 664 – 677: This paragraph is a little cumbersome to follow because it combines the rhythmicity result with the peak/trough way of quantifying the SWD firing rate changes. It is inconvenient having to switch back and forth between fig 3 and suppl fig 2. I would suggest moving the “all” panels from suppl fig 2 to fig 3. Alternatively, the rhythmicity results could move to a separate paragraph, if you want to make it a separate point. But if its purpose is basically to set up the peak/trough quantification (and the point that the trough deficit is greater than the peak surplus), then it’s fine. Also, the reader should be reminded of the timescales (or frequency range) on which you compute rhythmicity. You do mention your definition of rhythmicity in line 724, but I think it should be referenced here as well (or instead).

AUTHOR RESPONSE:

Thank you for these suggestions, we agree that the paragraph was cumbersome previously and have made the following changes: rhythmicity and peak/trough analysis are in separate paragraphs (because the former is independently worth emphasizing); the “all” groups have been moved from suppl Fig. 2 to a new main Fig. 4 together with the original Fig. 4 D-G. We have also added a line regarding the method and time bins we used to calculate rhythmicity (line 745): “Specifically, we defined rhythmicity as the inverse of the coefficient of variation of inter-spike intervals, calculated in 2 s time bins.”

REVIEWER COMMENTS:

Line 702: delete “or”

This and the following paragraphs: After you define SD, SI, OP, and NC you should consistently use only the abbreviations throughout the rest of the manuscript.

AUTHOR RESPONSE:

Thank you for spotting these issues, we have adjusted accordingly.

REVIEWER COMMENTS:

Lines 748-767: For this paragraph I have the following comments/concerns:

1) I don’t think, based on what most readers would consider an appropriate level of mechanistic insight, that you can claim to have identified a mechanism for the transient increase of the OP group. In my opinion that would require you to show e.g. that OP neurons normally receive strong inhibitory input from some class of neurons, and because those specific interneurons are suppressed during this time frame, OP neurons are hyperactive, but then a rebound excitation from another subpopulation stops this excitation, or something along those lines. The analysis you are showing is relevant in that it explains how your peak/trough calculations give you the characteristic OP behavior, which may hint at potential mechanisms behind that behavior, but it does not reveal the underlying circuit or cellular mechanism itself. Please adjust the language accordingly, or provide to the reader your exact definition of “mechanism” as used in this context.

2) I would add appropriate plots depicting the statistics you mention throughout lines 754-762 in figure 4.

3) Figure 4 c) at first glance looks like it is still aligned with the groups in b), which is a bit confusing. If possible, make it clearer that these panels do not match the categories above or arrange them differently.

4) While it makes sense that the high peak surplus in the first second plays a major role in separating the OP neurons from the other groups, I don't understand how the oscillation frequency plays a role. The EEG waveform presumably changes from the first second to the rest of each SWD in the same way, right? So are the OP neurons more strongly coupled to the EEG spikes than the other groups? Otherwise their oscillation frequency elevation in the first second should not be stronger than the other groups, right? Please explain and make adjustments to the text in this section.

AUTHOR RESPONSES:

1) Thank you for these comments and suggestions, we have adjusted the language used in this paragraph to make it clear that we have not tested or revealed a causal role for the OP neurons and the patterns in their firing. Specifically, we have replaced "mechanisms of" with "temporal characteristics that could explain the transient increased firing around the time of SWD onset in the OP subgroup" from line 872.

2) These plots have now been added to a new Supplementary Figure 5 as Figure 4 was very full already. The text in the Results lines 875 has been modified to refer to these new plots.

3) We have bolded the headings on each plot in Fig. 4C and added "OP" to make it clear that the panels do not match the categories in A and B.

4) We have added text clarifying that the higher initial frequency is a global property of SWDs in the Results from line 880, and that the greater initial peak/trough ratio of OP neurons is required to explain the onset peak: "However, this alone would not explain the unique transient peak feature in these neurons, as the higher first-second SWD frequency should affect the firing of all neurons not just OP neurons. Therefore we investigated the intensity of firing of OP neurons during peak and trough periods in the first second of seizures."

REVIEWER COMMENT:

Line 771: Please provide a reference.

AUTHOR RESPONSE:

Apologies, there should have been references here. They are now added

REVIEWER COMMENT:

Line 783: Please add statistical significance and % values.

AUTHOR RESPONSE:

Thank you for catching this oversight, statistical significance and magnitude are now added.

REVIEWER COMMENT:

Lines 787-795: Unless you do not have the data or you feel that it would be completely outside the scope of the study to run this analysis, it would be very intriguing to see how the different SWD-participating subgroups changed over these long pre-seizure periods. This seems like a

very obvious piece of information to add, and it would make a useful additional supplementary figure/panel. (optional)

AUTHOR RESPONSE:

While we fully agree that this information would be very interesting, when divided into the subgroups our sample size is not sufficient to overcome the greater variability introduced over long pre-seizure periods and so the data are too noisy to be informative. We hope to incorporate a pre-seizure state analysis into our future further characterization of the subgroups.

REVIEWER COMMENT:

Lines 799-802: I would suggest briefly relating your overall findings to previous studies that have looked at similar questions in humans and rodents, either here or in the following paragraph.

AUTHOR RESPONSE:

Thank you for the suggestion, we have added a brief statement highlighting the primary contrasts with previous rodent and human studies from line 975: “Prior work has not successfully replicated human fMRI and behavioral changes during absence seizures in an animal model, nor has it elucidated the neuronal activity patterns or the novel neuronal subtypes contributing to these changes.” Additional relation of our results to numerous previous studies in humans and rodents is provided in subsequent paragraphs throughout the remainder of the Discussion, to place this work in context of prior cited studies.

REVIEWER COMMENTS:

Line 808: Please add a reference.

Line 809: Please elaborate: “those seen in human absence epilepsy”, such as....? [Refs]

Line 816: Please add a reference. Also, please mention other previous studies that have shown neuronal suppression during SWDs. While it is interesting to juxtapose this suppression with the opposite phenomenon of hyperactivity in most other epileptic seizure types, I don't think you should omit previous results here that have reported suppression during SWDs, whether in GAERS or other rodent model organisms.

Lines 823-832: Please add more information on how your results compare to previous studies looking at single unit activity in GAERS cortex, such as [Polack et al, J Neurosci, 2007] , [Jarre et al, Cerebral cortex 2017]

AUTHOR RESPONSE:

We have added references and greater contextualisation with existing studies to the initial paragraphs of the Discussion. We have kept this quite brief here but it is discussed in greater detail later in this section.

REVIEWER COMMENT:

Line 834: I agree that this would be an important next step, but isn't it possible to at least separate pyramidal from GABAergic units based on extracellular waveforms? Unless you feel

that this is completely outside the scope of this study, this would be an important component to add, even just as a supplementary figure. (optional) If not, this possibility or how to employ other suitable methods should be discussed in more detail.

AUTHOR RESPONSE:

Separating excitatory from inhibitory neurons would absolutely be the first step in characterizing these subgroups, but from our dataset we are not able to confidently attribute these identities based on the waveform alone. Although previous studies do suggest that broader waveforms are excitatory and narrower inhibitory this does not seem to be a fully-established relationship, and we don't have a large enough population of simultaneously-recorded neurons to be able to validate their identities based on their cross-correlation with other synaptically-connected cells (e.g. see Henze et al. J Neurophysiol 2000, Becchetti et al. Front Neural Circuits 2012). We have added lines suggesting the first steps for characterization from line 1032: "Relevant approaches may include juxtacellular recordings with labelling, and simultaneous recording of paired neurons to assign excitatory and inhibitory identities^{64, 65}".

REVIEWER COMMENT:

Paragraph starting at line 860: Please discuss how your observations and proposed future directions might fit with recent studies showing that modulation of neuronal subtypes, such as certain interneurons, can suppress SWDs in rodent models, e.g. in [Panthi & Leitch, Front cell neurosci 2021] or [Clemente-Perez et al, Cell Rep 2017]. Moreover, as mentioned above this section would benefit from an analysis of the four subgroups in terms of their firing patterns during the 60 sec pre-SWD periods. Does the decrease in gamma EEG power point towards a specific class of neurons that are hypoactive? Another point to expand a bit here would be what the roles of the four subgroups of SWD -modulated neurons in the suppression of conscious experience/behavior might be. Do any of these groups coincide more strongly with the behavioral arrest you measured?

AUTHOR RESPONSE:

We thank the reviewer for the excellent suggestion to incorporate previous demonstrations of subtype-specific interventions in SWDs, we have now done so in the Discussion from line 1049: "Excitingly, suppression of rodent SWDs with cell-type specific interventions has been demonstrated^{68, 69}, and our results may provide an avenue to establishing the behavioral translatability of such interventions. In this case the populations we have observed must first be characterized before they may be targeted for selective modulation." As already noted, we were unable to relate a specific neuronal subclass to state changes due to sample size, and we do not have simultaneous neuronal and behavioural data so we cannot at this point investigate the relationship between the subgroups and behavioural arrest, but these will be critical future directions. We have now added the following to the Discussion from line 1059: "In addition, if a specific neuronal subtype could be associated with the state changes preceding seizures, or with the impairment of consciousness, those neurons could be specifically targeted to prevent these pathological features."

REVIEWER COMMENT:

Line 922: Please add a reference.

AUTHOR RESPONSE:

We apologize for this oversight, a reference has been added.

REVIEWER COMMENT:

Paragraph starting at line 931: I think [Polack et al, Cerebral cortex 2009] also recorded single unit activity from different thalamic nuclei in anesthetized GAERS, you might want to reference this study for comparison. I suppose you could easily repeat your experiments with anesthesia to test your hypothesis that being awake may be a reason for the dissociation between fMRI and e-phys measurements. (optional). Presumably this is outside the scope of the current study, but it should be discussed.

AUTHOR RESPONSE:

We agree that this would be a very interesting avenue of research and would be very informative regarding the effects of anesthesia on SWDs, on neurovascular coupling, etc. We do also feel it is outside the scope of this study and have added a statement from line 1155 suggesting it as an avenue for future research: “A future investigation of BOLD fMRI and neuronal activity in anesthetized GAERS may, by contrast with our current results, help to elucidate the exact effects of anesthesia on the neural and hemodynamic correlates of SWDs.”

Reviewer #2 (Remarks to the Author):

REVIEWER COMMENT:

This submission by McCafferty et al. is exceptionally significant to the fields of neurobiology of disease, neuroscience of consciousness, and thalamocortical network physiology. Their paper addresses two key problems concerning the understanding of the neurophysiology of absence seizures: 1) the apparent discrepancies between animal absence seizure models regarding cortical activation and consciousness impairment and 2) a mechanistic relationship between cortical and thalamic individual neuronal firing to seizure development and evolution. Importantly, the investigators then successfully link their single neuron activity results to the observed consciousness impairment and cortical activation.

Their experimental methodology is state-of-the-art, sound, and appropriate for the questions asked. In contrast to previous studies, they developed methods for performing the behavioral tests without overly alerting the subjects and accomplished the fMRI and electrophysiological studies without anesthesia, techniques that are crucial for the study of absence seizures and that will undoubtedly be used by others. They applied the expected analyses of their fMRI, behavior, and electrophysiological data, used appropriate statistical tests, and formed conclusions justified by the data. The methods are well-written and provide enough detail to be reproduced (except for some minor suggestions outlined below).

AUTHOR RESPONSE:

We sincerely thank the reviewer for their careful reading of our manuscript and for the valuable feedback that they have provided.

REVIEWER COMMENT:

Minor points:

1. Methods: They stated they recorded from the somatosensory cortex representing the trunk (S1TR) rather than the typical perioral region "to avoid extreme values in facial area." What is meant by this? Were they not able to isolate single units from the perioral region?

AUTHOR RESPONSE:

We have now clarified our reasoning for avoiding this region from line 721: "Specifically, deep layer peri-oral somatosensory neurons appear to initiate SWDs in GAERS, and this region of cortex temporally leads SWD activity in the related WAG/Rij model. Because our focus was on general cortical activity rather than on focal seizure initiation, we therefore recorded from subgranular layers (0.9 to 2 mm below pial surface) of somatosensory cortex in the trunk region (Supplementary Fig. 3)"

REVIEWER COMMENT:

2. Methods/Results: What cortical layers (supragranular, granular, infragranular) were recorded? It would be helpful to have an example of the histology for both cortex and thalamus in results (possibly in supplemental data).

AUTHOR RESPONSE:

We targeted Layer IV of cortex and moved gradually downwards using our microdrive over the days of recording. We have added example histological sections for cortex (showing the range of probe locations) and thalamus to a new Supplementary Figure 3. Please see the quoted addition in the previous response for layer details.

REVIEWER COMMENT:

3. Methods The diagram of the frame Supplementary 1 is difficult to understand – consider adding a picture of the mouse to the frame.

AUTHOR RESPONSE:

We have added an indication of the headplate's positioning to part B of this figure to help clarify the orientation and placement of the headplate.

REVIEWER COMMENT:

4. Methods Their use of female rats is justified for experimental reasons. I believe the journal requires the abstract to state the sex of the subjects when animals of only one sex were used.

AUTHOR RESPONSE: We thank the reviewer for this point, the sex of the animals is now stated in the abstract (line 39).

REVIEWER COMMENT:

5. Results – can their spike sorting differentiate interneurons from principal neurons? If so, do they cluster within their defined neuronal types?

AUTHOR RESPONSE:

Although it would be interesting we cannot reliably differentiate interneurons and principal neurons with our recordings. Previous recordings have used width of the spike waveform to make that distinction, but it is advisable to use paired monosynaptically interacting neurons to observe excitatory/inhibitory interactions on the cross-correlogram to validate the waveforms characteristics in each population, or histological identification of neurotransmitters in single juxtacellularly recorded neurons. We have added text recommending this as a future direction from line 1023: “Further work is needed to histologically and functionally identify these distinct neuronal types, which was not possible with the present silicon probe multi-contact recordings. Relevant approaches may include juxtacellular recordings with labelling, and simultaneous recording of paired neurons to assign excitatory and inhibitory identities^{64, 65}”.

REVIEWER COMMENT

6. Results – can they determine the timing of spiking in the cortex vs. thalamus – e.g. does cortex precede thalamus as was observed in other studies?

AUTHOR RESPONSE: as the cortical and thalamic neurons for this study were recorded separately we cannot tell whether one precedes the other with the degree of precision that would be useful in this circumstance.

REVIEWER COMMENT:

7. Results Figure 3B Why are the spikes separated by PC7 & PC10 rather than PC1-3?

AUTHOR RESPONSE: We thank the reviewer for the opportunity to clarify this point. There are 3 principal components derived from each channel in this sorting method, with the channels grouped in 4s or 8s. The first three PCs of channels 1, 2, 3 and 4 are respectively, PC1-3, PC4-6, PC7-9, and PC10-12. Therefore PC7 and PC10 are the first principal components from channels 3 and 4 in this group, respectively. Furthermore, all principal components computed are used for clustering but we chose to represent PC7 and PC10 in this case as the dimensions showing clearest separation between clusters. We have added the following text to the caption of Fig. 3B to make this more clear: “PC7 and PC10 represent the first principal components from two channels in this example that show clear separation between clusters.”

REVIEWER COMMENT:

8. Results Figure 3C shows broad band recordings – presumably to highlight the transition from pre-ictal to ictal EEG. However, broad band does not demonstrate the unit recordings. Possibly an inset with the high frequency recordings would be helpful.

AUTHOR RESPONSE:

Thank you for this suggestion. We have changed the colour of the spikes in the insets of the figure (now Fig 3, part D) in order to highlight the successful unit recording, and we added a new part C with the high frequency component of the recording as suggested.

REVIEWER COMMENT:

9. Typographical errors:

- a. Fig 3D legend “2-3” (omitted word seconds)
- b. Line 828 more numerous than SI neurons (should be “than” and not “that”)

AUTHOR RESPONSE:

Thank you for these corrections, they have now been made.

Reviewer #3 (Remarks to the Author):

REVIEWER COMMENT:

The manuscript by McCafferty et al. explore the neuronal basis of impaired consciousness in absence seizures. They used a well-accepted spontaneous rat model for human absence epilepsy and combined EEG recordings with behavioural measures and fMRI of awake, restrained-habituated GAERS rats. The manuscript is well-written. Methods seem well-established and are described in detail. Overall conclusions are sound. I would recommend publication of this paper. The authors convincingly present behavioural loss of consciousness, SWDs on EEG and decreased cortical fMRI signal in agreement with findings in human absence epilepsy. On neuronal level, they identified decreased overall cortical and thalamic firing and increased rhythmicity. Neuronal sorting revealed four distinct patterns of neuronal activity changes, with the biggest subgroup showing sustained decrease of firing.

AUTHOR RESPONSE:

We greatly appreciate the reviewer’s careful reading of our manuscript and their positive and constructive feedback.

REVIEWER COMMENT:

Further characterization of the identified neuronal subpopulations (histology and function) has not been performed in this study. This finding is indeed interesting and warrants further investigations. For now, speculating about using these different neuronal groups as targets for future therapies seems a bit far-reaching, without any further knowledge about respective cell specifics (end of Intro and discussion line 851ff). Based on their experience in the field, could the authors suggest any potential candidates?

AUTHOR COMMENT:

We fully agree with the reviewer’s point and have now endeavoured to make it clear that the first priority would be to find out the identities and causal roles of the cells in question, with therapeutics a long way down the line. We have removed the speculation from the end of the introduction and added text from line 1008 (“Excitingly, suppression of rodent SWDs with cell-type specific interventions has been demonstrated^{68, 69}, and our results may provide an avenue to establishing the behavioral translatability of such interventions. In this case the populations we have observed must first be characterized before they may be targeted for selective modulation.”), as well as clarifying in line 1116 that these therapeutic targets are in the “long-term”..

REVIEWER COMMENT:

The authors established fMRI in awake, restrain-habituated animals in their study. They describe the habituation procedure in detail. The authors are obviously aware of and state that seizures are

more likely to occur in a state of relaxed wakefulness. To overcome the issue that SWD may be interrupted due to loud noise or too stressful/demanding stimuli they have developed a closed-loop test of auditory responsiveness by decreasing stimulus intensity and increasing inter-stimulus interval durations for the behavioural part of the study. I am wondering how stress and a higher state of arousal due to restraint and to the unavoidable loud noise during MR scanning has affected the occurrence and duration of SWDs in their experiments. Also, Isoflurane used for positioning of the animals will temporarily suppress SWDs. Specifically, more detailed information regarding when first SWDs are usually observed during a MRI examination, how many SWDs are observed and the mean duration of SWDs in comparison to the behavioral cohort are of interest. This information can not fully be retrieved from the 90 sec exemplary simultaneous EEG recording (in suppl fig 1) nor from the notion that 670 SWDs were observed in 18 animals (in fig 1). Could the authors identify potential differences in baseline arousal state between MRI and behavioural cohort or exclude such by comparing baseline spectrograms?

AUTHOR RESPONSE:

We thank the author for this valuable suggestion, and we have now added details on when first SWDs are usually observed during a MRI examination, how many SWDs are observed and the mean duration of SWDs during fMRI recordings from line 258: “SWDs began to appear after a mean of 25 ± 1.8 minutes, and there were an average of 28.5 ± 3 SWDs per recording lasting on average 15.1 ± 1.7 seconds. The corresponding values for the behavioral recordings are first SWD at 30.3 ± 1.8 minutes, SWD rate of 60.5 ± 2.8 per hour, and SWD duration of 8.7 ± 0.2 seconds.” We can see therefore that the time to first onset and total time spent in SWD is very similar between the two types of recording, with fMRI SWDs tending to be longer and less frequent. We agree that there may be differences in arousal state during the MRI experiments compared to behavioral and neuronal recordings. Unfortunately we cannot usefully compare spectrograms from the fMRI recordings with other spectrograms as scanner artifacts produce significant noise in relevant bands. However, we did evaluate the amount of spontaneous sleep activity, where we found a similar very low amount of sleep for both fMRI and behavioral testing recordings (where animals were presumably relatively engaged or aroused) in comparison to the neuronal recordings where sleep was more often observed. We now include this information in the Methods from line 416: “Periods of sleep were identified based on sharp increases in the 1-4 Hz frequency band, and excluded from analysis (note that these periods occurred only during neuronal activity recordings, and not during fMRI or behavioral recordings, presumably due to the more stimulating sensory environment in the latter two protocols).” We acknowledge that arousal state is very likely different between the fMRI and the other cohorts. However, at least the arousal state in the present fMRI experiments is more similar to awake behaving experiments than the state in previous fMRI studies done in anesthetized or sedated animals. To highlight the importance of arousal state among other factors for interpreting our fMRI results in comparison to prior studies, we have added the following to the Discussion from line 1128: “We suggest that the contrast with previous animal model SWD hemodynamics is due to the lack of anesthetic or sedative agents in our preparation, and consequently the different baseline arousal states, levels of neural activity, and neurovascular demand.”

REVIEWER COMMENT:

For behavioural training animals were food restricted. The authors state that they used the initial body mass as reference value. Female rats between 3 and 7 month of age are not fully matured

and under a normal ad libitum diet will gain body weight. I guess behavioral experiments lasted in total weeks (from first training session to termination). From an animal welfare perspective, it would be more reasonable to compare body weights to not food-restricted, age-matched animals.

AUTHOR RESPONSE:

We agree that this is an important consideration. The reviewer is correct that the food-restricted training period lasted a maximum of 3 weeks, and as noted from line 274 it was ended before surgery and was not re-started: “This 90% target mass was maintained through training and food restriction ceased approximately 1 week before implantation to allow recovery for surgery.” Therefore from an animal welfare perspective we believe that none of the animals suffered excessive food restriction, however we agree that comparison to age-matched non-food restricted animals certainly would be appropriate if longer-term food restriction were to be employed.

REVIEWER COMMENT:

To investigate temporal dynamics of fMRI associated with SWD the authors set the pre-seizure baseline = 5 seconds immediately before seizure. EEG recordings revealed a rapid dip in neuronal firing 2-3 sec before SWD initiation (figure 3D and E). How would shifting the onset/pre-seizure baseline affect the outcome of GLM and time course analysis?

Similarly, did the authors try to use the time point of the steep power shift in the high and low frequency band 40-60 sec before SWD onset as a regressor for GLM or take it into account for definition of the pre-seizure baseline in the time course plots?

AUTHOR RESPONSE:

We thank the reviewer for these interesting points. We have shifted the time course analysis to use the 5 second window from 8 seconds to 3 seconds before SWD initiation as the baseline, and reanalyzed our timecourse data. This produced very similar results to the analysis using the 5 to 0 baseline window, as now shown in the new plots in Supplementary Fig. 1E. We also describe this analysis in the manuscript from line 478: “Because we observed decreased firing in the ~3 seconds before seizure initiation (see Fig 4 A, C), we re-analyzed the fMRI timecourses with an adjusted 5 s baseline (8 seconds to 3 seconds pre-seizure) but did not find any apparent differences in the results (Supplementary Fig. 1E).” The magnitude of the changes using the adjusted baseline also did not quantitatively change by a substantial amount – for example, the mean decrease in fMRI signal in the somatosensory cortex during SWDs with the original baseline (Figure 1C) was 0.97 ± 0.18 % SEM, whereas with the adjusted baseline (Supplementary Fig. 1E) it was 1.01 ± 0.17 %. Similarly, the mean fMRI increase in the ventrobasal thalamus with the original baseline was 0.51 ± 0.11 %, and with the adjusted baseline it was 0.49 ± 0.11 %. As for the GLM analysis, because it uses as baseline all non-seizure periods throughout the run rather than just the 5 seconds preceding SWDs, we expect the GLM analysis to be even less sensitive to excluding or including the 3 seconds prior to SWDs. Regarding the EEG power shifts 40-60 sec before SWDs, unfortunately we would not be able to use the longer-term pre-seizure changes in our fMRI analyses because the frequency ranges of these subtle EEG power shifts are largely obscured by MRI scanner artifact.

REVIEWER COMMENTS:

Minor comments

Methods, line 164 ff - Please provide information about analgesia

Results line 702 – remove “or” after increase

AUTHOR RESPONSE:

These issues have now been addressed.

We thank the reviewers again for their helpful comments and suggestions. We feel that these revisions have significantly strengthened the manuscript, and hope that it is now acceptable for publication. Thank you very much for your kind consideration.

Sincerely yours,

Hal Blumenfeld, MD, PhD

Mark Loughridge and Michele Williams Professor
Depts. of Neurology, Neuroscience, Neurosurgery
Director, Clinical Neuroscience Imaging Center
Yale School of Medicine

Cian McCafferty

Dept of Anatomy & Neuroscience
University College Cork

REVIEWERS' COMMENTS

Reviewer #1 (Remarks to the Author):

I would like to thank the authors for their excellent work revising the present manuscript and their detailed responses to my questions and concerns. I believe that the quality of their work was already high to begin with, and the presentation of the results is now equally impressive. I had just a handful of minor additional suggestions I would like the authors to consider, which should not take a lot of time to implement (see attached file). I am confident that this publication will attract a great deal of attention from the audience, and I hope to see further research from the authors into important questions raised by this study!

Reviewer #2 (Remarks to the Author):

The authors have answered all my questions. I recommend acceptance of the manuscript.

Reviewer #3 (Remarks to the Author):

The authors have addressed all comments satisfactorily and adjusted the manuscript accordingly. I am supporting the decision to publish the manuscript.

Yale University
School of Medicine

HAL BLUMENFELD, MD, PHD

Mark Loughridge and Michele Williams Professor
Director, Yale Clinical Neuroscience Imaging Center
Professor of Neurology, Neuroscience, Neurosurgery
Epilepsy Service
Yale University School of Medicine
New Haven, CT 06520-8018
Tel: (203) 785-3865
Fax:: (203) 737-2538
E-mail: hal.blumenfeld@yale.edu

RE: NCOMMS-22-25827B

Nov 11, 2022

Dear Reviewers,

Thank you again for your contributions to this manuscript. Please see the below response to each of your points, and the associated changes in the revised manuscript.

Reviewer #1:

REVIEWER COMMENT:

You rightfully start by saying we need to understand how neuronal activity *causes* impaired consciousness in seizures, so to be consistent with that directionality I would change the second part of the sentence accordingly... I would say something like "reflect" instead of "affect", and "contribute to absences via" instead of "display". I think otherwise the reader might wonder whether manipulating the activity of the different neuronal subtypes you identified to potentially prevent seizures could actually work, or if their activity patterns are a symptom rather than a cause of the seizures.

AUTHOR RESPONSE:

We have made the two suggested edits and agree that they improve the clarity of our intended messages.

REVIEWER COMMENT:

To add more significance to these findings, I would consider adding references here such as the ones you cite later in the discussion that show evidence of pre-ictal changes on longer time scales in human studies.

AUTHOR RESPONSE:

We have added the suggested references.

REVIEWER COMMENT:

Please add the range of mean +/- SD of #units per animal.

AUTHOR RESPONSE:

We have added these values.

REVIEWER COMMENT:

BOLD abbr. should be introduced in the previous paragraph.

AUTHOR RESPONSE:

We have adjusted appropriately.

REVIEWER COMMENT:

Please remove reference 62 "Psychological functions during spike-wave discharges" because it is not relevant. Keep reference 60 "The relationship between generalized and paroxysmal EEG discharges..." because it does show effects of arousal state on SWDs. Could also add the following reference, which was the first observation that increased arousal state suppresses absence seizures:

Lennox WG, Gibbs FA, Gibbs EL. Effect on the electroencephalogram of drugs and conditions which influence seizures. Arch Neurol Psychiat 36, 1236-1250 (1936).

AUTHOR RESPONSE:

We have removed the indicated reference.

We hope that these revisions meet your expectations,

Sincerely yours,

Hal Blumenfeld, MD, PhD

Mark Loughridge and Michele Williams Professor
Depts. of Neurology, Neuroscience, Neurosurgery
Director, Clinical Neuroscience Imaging Center
Yale School of Medicine

Cian McCafferty

Dept of Anatomy & Neuroscience
University College Cork